



# CBRA: The first multi-annual (2016-2021) and high-resolution (2.5 m) building rooftop area dataset in China derived with Super-resolution Segmentation from Sentinel-2 imagery

Zeping Liu[1,2], Hong Tang[1,2,3], Lin Feng[2], Siqing Lyu[2]

[1]Key Laboratory of Environmental Change and Natural Disaster of Ministry of Education, Beijing Normal University, Beijing 100875, P. R. China

[2]Beijing Key Laboratory for Remote Sensing of Environment and Digital Cities, Faculty of Geographical Science, Beijing Normal University, Beijing 100875, P. R. China

[3]State Key Laboratory of Remote Sensing Science, Faculty of Geographical Science, Beijing Normal University, Beijing 100875, P. R. China

*Correspondence to*: Hong Tang (hongtang@bnu.edu.cn)

**Abstract.** Large-scale and multi-annual maps of building rooftop area (BRA) are crucial for addressing policy decisions and sustainable development. In addition, as a fine-grained indicator of human activities, BRA could contribute to urban planning and energy modelling to provide benefits to human well-being. However, it is still challenging to produce large-scale BRA due to the rather tiny size of individual buildings. From the viewpoint of classification methods, conventional approaches utilize high-resolution aerial images (metric or sub-metric resolution) to map BRA; unfortunately, high-resolution imagery is both infrequently captured and expensive to purchase, making the BRA mapping costly and inadequate over a consistent spatio-temporal scale. From the viewpoint of learning strategies, there is a non-trivial gap that persists between the limited training references and the applications over geospatial variations. Despite the difficulties, existing large-scale BRA datasets, such as those from Microsoft or Google, do not include China, hence there are no full-coverage maps of BRA in China yet. In this paper, we first propose a deep-learning method, named Spatio-Temporal aware Super-Resolution Segmentation framework (STSR-Seg) to achieve robust super-resolution BRA extraction from relatively low-resolution imagery over a large geographic space. Then, we produce the multi-annual China building rooftop area dataset (CBRA) with 2.5 m resolution from 2016-2021 Sentinel-2 images. The CBRA is the first full-coverage and multi-annual BRA data in China. With the designed training sample generation algorithms and the spatio-temporal aware learning strategies, the CBRA achieves good performance with the F1 score of 62.55% (+10.61% compared with the previous BRA data in China) based on 250,000 testing samples in urban areas, and the recall of 78.94% based on 30,000 testing samples in rural areas. Temporal analysis shows good performance consistency over years and the well agreement to other multi-annual impervious surface area datasets. The STSR-Seg will enable low-cost, dynamic and large-scale BRA mapping (https://github.com/zpl99/STSR-Seg). The CBRA will foster the development of BRA mapping and therefore provide basic data for sustainable research (Liu et al., 2023; https://doi.org/10.5281/zenodo.7500612).



## 1 Introduction

The building rooftop area is an essential indicator of human activity (Huang et al., 2021a), sustainable urbanization (Appolloni et al., 2021; Burke et al., 2021), building energy modeling (Byrne et al., 2015; Chen et al., 2022), urban planning (Nadal et al.,

2017) and quick response to natural disasters (Chen et al., 2022; Ge et al., 2023) in the recent years. Such dataset has thus become pivotal in a range of policy decisions by the government, such as arranging the correlation between economic development and demographic growth, and how and where to implement public service. However, many regions might lack the kind of information to systematically assess this development in both large geographical regions and long time periods (Burke et al., 2021). In the meantime, satellite remote sensing has been the prominent measure for urban mapping of our earth

(Zhu et al., 2022b), especially in developing regions where survey data or human-labeled data is rather difficult to obtain (Ayush et al., 2021a). Compared to the traditional survey-based methods (Kuthanazhi et al., 2016; Jones and Hobbs, 2021), remote sensing could observe large areas at a potentially low cost, thus allowing tracking of the building dynamic of developing regions.

Unlike other datasets containing building information from satellite imagery, such as Impervious Surface Area (ISA) or

Human Settlement Footprint (HSF), the Building Rooftop Area (BRA) requires a higher spatial resolution for well identification due to the tiny size of objects of interest (e.g., residential houses). Typically, the ISA (Zhang et al., 2022a; Huang et al., 2022) or HSF (Marconcini et al., 2020a; Qiu et al., 2020) are derived from the imagery with a spatial resolution of decametric level (e.g., 30 m or 10 m). While the BRA (Liu et al., 2022; Zhang et al., 2022b) utilizes high-resolution aerial imagery with a resolution of metric level (e.g., 1 m). However, high-resolution aerial imagery is costly and potentially not

publicly available. For example, the price of WorldView-2 is $23/km$^2$ (HR Imagery Ordering, 2022), denoting mapping one city at 1 m requires at least $115,000. The high data expenditure makes large-scale BRA possible only for large companies, e.g., Google and Microsoft, which have implemented the continental-scale BRA of Africa (Sirko et al., 2021) and global BRA (GlobalMLBuildingFootprints, 2022) using Google Map and Bing Map, respectively. To tackle it, many international endeavors analyze building rooftops by utilizing open-access Google Earth Satellite (GES) images (Liang et al., 2018). Most

recently, Zhang et al. (2022b) utilize GES imagery and obtained 90-cities-BRA for China under the resolution of 1 m. Due to the uneven distribution of GES image patches, as well as the inconsistent geometric offset and acquisition time (e.g., the image is obtained from various satellite sensors with different acquisition times), the existing BRA has geospatial inconsistency in practical applications, thus limiting the generalization to the questions of broad social importance, particularly in large geographic and time-scale mapping (further discussion in Sect. 2.2).

China is a rapidly developing country, with 4.3% urbanization growth in the past five years. According to the National Bureau of Statistics of China, the urbanization rate of China has reached 64.72% in 2021 but the rural population is still large, accounting for 509.79 million people. The "dual-track" society structure indicates that human activity occurs variously in both developed and developing regions of China (Guan et al., 2018a). The existing large-scale BRA dataset provided by Microsoft and Google do not include China, while the BRA produced by Zhang et al. (2022b) only covers 90 cities in China. In addition,



to our best knowledge, few of the existing BRA provide multi-annul results, and such temporal information is of great significance to developing countries such as China.

To foster the development of the observation of human living space, and to provide all stakeholders with free access to data to monitor building rooftop dynamic at a national scale and high spatio-temporal resolution, we introduce the CBRA (China Building Rooftop Area) dataset, which reports the pixel-level building rooftops distribution along with their dynamic, from

2016 to 2021, on a national scale. The CBRA is derived from the Sentinel-2 imagery (up to 10 m spatial resolution). To meet the spatial resolution of the BRA needs and tackle the lack of reliable training references, we propose a deep-learning-based framework, called the Spatio-Temporal aware Super-Resolution SEGmentation framework (STSR-Seg). The STSR-Seg could capture the high-resolution context from the Sentinel-2 imagery and the low-resolution land cover data, thus achieving robust spatio-temporal results of BRA at 2.5 m resolution. With the proposed STSR-Seg, the CBRA outperforms the existing BRA

in the urban region of China, with overall accuracy and F1 score of 2.5 m, 82.85%, and 62.55%, respectively. The main contributions are as follows:

(1) The free access to the CBRA, which is the first multi-annual (2016-2021) and 2.5 m BRA products at a national scale (e.g., China). The CBRA is also the full-coverage BRA data in China, including both urban regions and rural regions.

(2) The CBRA is a spatio-temporal consistency dataset among the existing BRA datasets, only generated by Sentinel-2

satellite imagery with specific acquisition time and location.

(3) The proposed STSR-Seg framework could achieve robust spatio-temporal super-resolution output, thus reducing the data expenditure of both the high-resolution imagery and training references for the large-scale BRA applications

The remainder of this paper is arranged as follows: Sect. 2 reviews and analyses the background of our methodology and the building-related datasets. Sect.3 introduces the data we used for dataset generation. Sect. 4 describes the methodology in

detail. The following Sect. 5 provides results, evaluations, and analyses of the CBRA. Discussions are listed in Sect. 6. Finally, the conclusions are drawn in Sect. 7.

## 2 Background

To provide an overview of the involved methodology and dataset, Sect. 2.1 would briefly describe the methodological background. Moreover, the existing building-related products would be reviewed in Sect. 2.2.

### 2.1 Methodological background

Fig. 1 shows an overview of the background of the involved methods and their relations to our methodology. Specifically, we will focus on two fields of deep learning in earth observation, i.e., the super resolution and semantic segmentation classification methods, and the weakly-supervised learning algorithms.



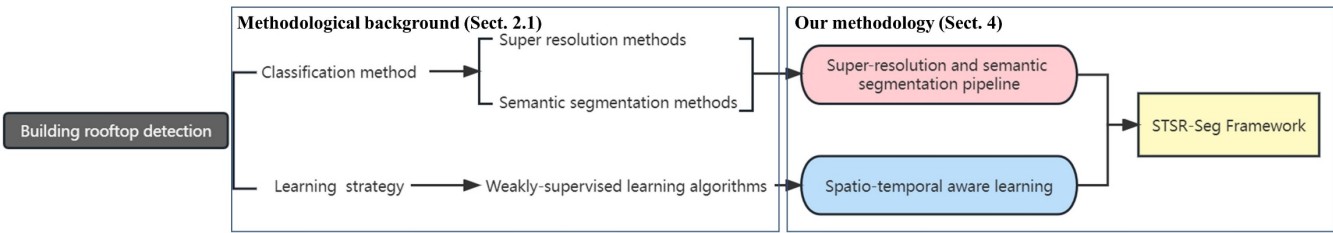

Figure 1: An overview of the methodological background and its relation to our proposed methodology.

### 2.1.1 Super resolution and semantic segmentation methods

The great success of deep convolutional neural networks (CNN) in the computer vision field has already revealed a new era for earth observation (Hoeser et al., 2022), like super resolution (SR) and semantic segmentation (SS). Utilizing the SR methods could transfer the low-resolution image to the high-resolution, thus expanding the cheaper satellite with a coarser resolution to the application demanding high-resolution data (Shermeyer and van Etten, 2019). He et al. (2021) utilize low-resolution and high-resolution image pairs to learn the SR model and map the low-resolution image to the high-resolution, while Xu et al. (2021) apply only the high-resolution label, achieving strong performance in the downstream high-resolution tasks. The SS, which is a pixel-wise classification task, also has a lot of applications in earth observation, such as land use mapping (Zhu et al., 2022a) and disaster detection (Munawar et al., 2022). Recently, the SR and SS are combined to realize high spatial resolution tasks, like building counting (He et al., 2022) and boat detection (Zhang et al., 2019). Such state-of-the-art (SOTA) SR and SS approaches have shown great accuracy in various benchmark datasets and competitions (Wang et al., 2022), and their huge potential in large-scale and time-series building rooftop mapping is ripe for discovery.

### 2.1.2 Weak-supervised learning algorithms

Remote sensing offers an enormous supply of data provided by the over 1000 satellites currently in orbit. Many downstream tasks, however, are limited by the lack of reliable annotations, which are particularly costly as they often require expert knowledge, or expensive ground sensors (Robinson et al., 2019; Manas et al., 2021). Besides, satellite imagery is various in both geography and time. Factors like season and climate pose great generalization challenges to the deep learning model, while these factors are difficult to be human-labelled and explicitly learned by the model.

Recent years have seen a proliferation of studies to tackle the above challenges, among which the weak-supervised learning algorithm has gained great attention in the earth observation field (Yue et al., 2022). One is the pre-text task learning algorithm. It is implemented by forcing the model to learn representations of other related tasks simultaneously, e.g., the coordinates of the input imagery (Muhtar et al., 2022), and the night-time light intensities (Xie et al., 2016). Another is the contrastive learning algorithm, which is to learn the representations by pulling positive (similar) feature pairs closely in latent space and pushing the negative (dissimilar) feature far away from the positive. For example, Manas et al. (2021) and Ayush et al. (2021b) denote the imagery of the same location but at different times are positive. While Yang and Ma. (2022) denote the patch in images with the same land cover class as positive and different types as negative. The intuition of the weak-supervised learning



algorithm is to learn the representation from other related tasks with easily accessible labelled data, or learn the latent invariance from the observed imagery itself, thus alleviating the limitation of annotations of the downstream tasks.

Due to spatio-temporal variations, there is a shortage of reliable annotations for national-scale and multi-annual building rooftop detection. In the meantime, information such as acquisition time, image location, and land cover data are plentiful in the community. Overall, there are two primary challenges, each with a possible solution:

(1) The lack of reliable building rooftop annotations, especially in rural areas, poses a weakly supervised problem – utilizing low-resolution land cover data as supervision, since they could provide the information about "where possibly have built".

(2) The different acquisition time of imagery makes the image of the same location but at a different time have different image style, posing a challenge for the model generalization – implementing the contrastive learning algorithm to make the model invariant for the temporal discrepancies.

Based on the above observation, we propose a novel framework (STSR-Seg), where we utilize the state-of-the-art (SOTA) SR and SS approach and the weak-supervised learning algorithm, to achieve robust building rooftop detection in China.

## 2.2 Building-related products

So far, there have been a lot of studies focused on human living space or the land surface cover from different scales. These studies also give information about buildings. Early efforts usually focus on using very low-resolution satellite data, e.g., Defense Meteorological Satellite Program (DMSP) and Moderate Resolution Imaging Spectroradiometer (MODIS) data, to produce the Land Use and land Cover Change (LUCC) data (including urban or built cover information) at 100 m to 1 km spatial resolution (Schneider et al., 2003; Tateishi et al., 2011). With the free availability of Landsat and Sentinel satellite data, as well as the powerful geospatial cloud platforms (e.g., Google Earth Engine, GEE), more international studies work towards mapping at a finer spatial resolution (e.g., decametric) over long periods and large geographic, and providing more detailed building-related products such as the Impervious Surface Area (ISA), and the Human Settlement Footprint (HSF). For LUCC, ISA and HSF, it is witnessed a series of global mapping efforts in recent years, such as FROM_GLC (Gong et al., 2013), GAIA (Gong et al., 2020a), GISA-10m (Huang et al., 2022), GHSL (Corbane et al., 2021) and WSF (Marconcini et al., 2020b). The spatial resolution of the products aforementioned ranges from 30 m to 10 m, and the period ranges from 40 years to only 1 year (Table 1). These data provide the built cover information, or the impervious surface information, and are frequently used to conduct building-related studies (Fox et al., 2019). However, due to the resolution gap, these data may contain errors when specific to individual buildings (Fig. 2), which has thus inspired the investigation of the Building Rooftop Area (BRA) that can describe the individual buildings.

However, BRA mapping remains challenging and is not well solved due to the tiny size of individual buildings. Typically, the BRA demands remote sensing images with a metric or sub-metric resolution. Purchasing these images needs a very high data expenditure; hence, large-scale BRA mapping is relatively hampered compared with other building-related data aforementioned. Currently, the open-access large-scale BRA data are from Google (Sirko et al., 2021) and Microsoft



(GlobalMLBuildingFootprints, 2022), due to the fact that these companies can afford the high cost of large-scale building mapping. They utilize deep learning (e.g., semantic segmentation) methods, high-resolution imagery from Google map or Bing map, as well as the building rooftop ground truths by human labelled, to achieve continent-scale mapping (e.g., Africa) and global mapping, respectively. Unfortunately, China is not included in their products.

Recently, Zhang et al, (2022b) apple GES images and semantic segmentation methods to detect building rooftops of 90

cities in China in the year 2020. However, the GES images are collected from various kinds of high-resolution satellites, and have two potential problems when applied to large-scale mapping: (1) inconsistent geographical offset, and (2) inconsistent acquisition time, which results in spatio-temporal inconsistency in the generated product. Also, the product from Zhang et al, (2022b) does not cover the living space of more than 36% of China's population, e.g., the rural area.

Moreover, China is undergoing rapid urbanization and a rural-urban demographic transition (Guan et al., 2018b). A single

year of building rooftop distribution may not be sufficient for the research about sustainable development. To our best knowledge, few of the existing BRA provide a multi-annual mapping on a large-scale (e.g., national), or in a developing region (e.g., rural). Therefore, there is an urgent for BRA over both a large-scale area and a specific time span, to support various fine-scale applications.

Overall, the large-scale BRA data is currently limited, especially in China. In addition, there are no both multi-annual and

large-scale BRA data freely available to the public (summarizes in Table 1). To this end, we present the CBRA (China Building Rooftop Area) dataset by using the proposed STSR-Seg deep learning method in this study, which is with 2.5 m spatial resolution and 1 y temporal resolution, ranging from 2016-2021.

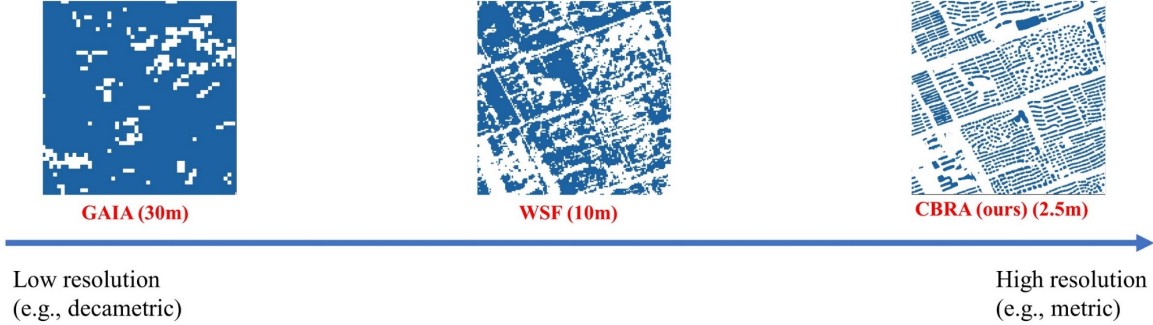


**Figure 2: An example of the result from several representative building-related datasets (121.344419° N, 31.093870° E). The GAIA (Gong et al., 2020b) reflects the impervious area (30 m). The WSF (Marconcini et al., 2020b) is a human settlement data (10 m). The CBRA (ours) is the building rooftop area data (2.5 m).**






**Table 1: The recent well-known building-related datasets and the existing large-scale BRA datasets.**

| Dataset | Data, scale, and time span | Resolution | Classification method and strategy | Type definition |
|---|---|---|---|---|
| FROM-GLC30 (Gong et al., 2013) | Landsat, global, 2015 | 30 m | Maximum likelihood classifier, random forests, and the support vector machine | LUCC data (including ISA) |
| GISA (Huang et al., 2021b) | Landsat, global, 1972-2019 | 30 m | Random forest classifier via hexagonal partitioning | ISA data |
| GAIA (Gong et al., 2020b) | Landsat, global, 1985-2018 | 30 m | An exclusion-inclusion approach | ISA data |
| WSF (Marconcini et al., 2020b) | Landsat 8 and Sentinel-1, global, 2015 | 10 m | Support vector machine | HSF data |
| GISA-10m (Huang et al., 2022) | Sentinel-1 and Sentinel-2, global, 2016 | 10 m | Random forest classifier via hexagonal partitioning | ISA data |
| GHSL (Corbane et al., 2021) | Sentinel-2, global, 2018 | 10 m | Convolutional neural networks with two-stage training | HSF data |
| Google BRA* (Sirko et al., 2021) | Google map, Africa, no specific time | 0.5 m | Semantic segmentation, pre-training, self-training, and polygonization | BRA data |
| Microsoft BRA* (GlobalMLBuildingFootprints, 2022) | Bing map, global (not cover China), no specific time | <1 m | Semantic segmentation and polygonization | BRA data |
| 90-cities-BRA (Zhang et al., 2022b) | Google Earth Satellite image, 90 cities in China, 2020 | 1 m | Semantic segmentation and vectorization | BRA data |
| CBRA (ours) | Sentinel-2, China, 2016-2021 | 2.5 m | Super-resolution segmentation and spatio-temporal aware learning | BRA data |

\* Results from Google and Microsoft are not specific in time, because the images they collected worldwide do not have consistent acquisition times.

## 3 Data

### 3.1 Satellite imagery

Sentinel-2 optical images are used for the CBRA mapping. The Sentinel-2 is an earth observation mission under the European Space Agency (ESA) Copernicus program, including a constellation of two satellites, i.e., Sentinel-2A and Sentinel-2B. The first Sentinel-2 satellite has observed the earth since June 2015, providing mainly four 10 m visible bands (i.e., RGB) and the near-infrared (NIR) bands, six 20 m short-wave infrared (SWIR) and red-edge bands, and three 60 m bands (Huang et al., 2022). In this paper, we only utilize the band with 10 m (i.e., RGB and NIR), since the previous study shows that introducing bands with coarser resolution potentially brings degradation in the performance of deep learning models (Adriano et al., 2021). After the testing and adjustment by ESA, Sentinel-2 has achieved complete coverage of China since 2016 (Huang et al., 2022). Therefore, we utilize the Level-1C top-of-atmosphere (TOA) reflectance product, which has been conducted with systematic radiometric calibration, geometric and terrain correction by ESA. To tackle the cloud noise, we utilize the Google earth engine (GEE) (Gorelick et al., 2017) to filter out the images with more than 20% clouds, and further conduct clouds and shadow removal by the quality bands to get cloud-free pixels. Finally, we perform median-composition of the filtered images within 1-year intervals. The number of image tiles for median composition (cloud under 20%) over China from 2016-2021 are shown

in Fig. 3. Note that there are several missing images in parts of southwest China. However, there are few human activities in
these regions, thus the impact on our results is negligible (Table. B1).

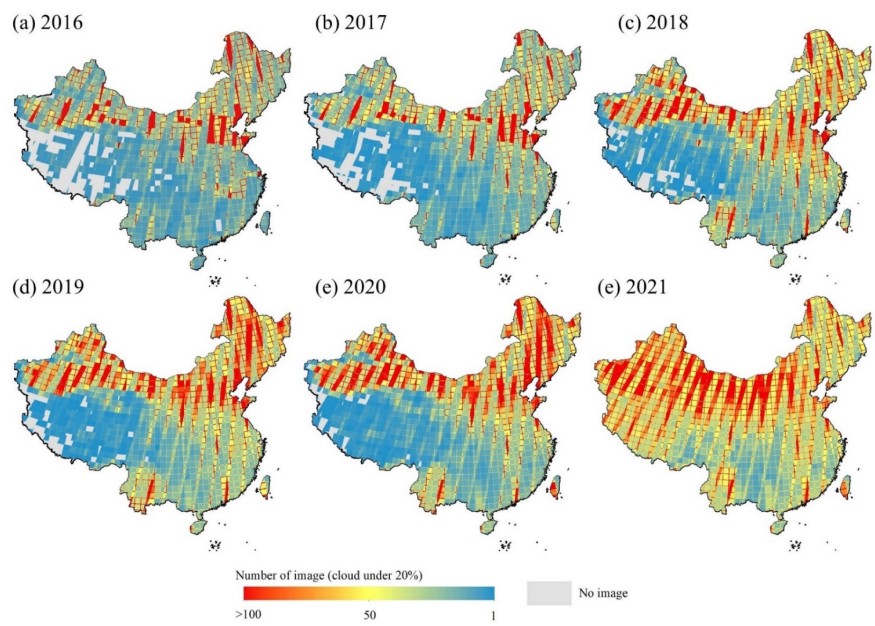

**Figure 3: Distribution of the Sentinel-2 images (cloud under 20%). Base map © OpenStreetMap contributors 2023. Distributed under the Open Data Commons Open Database License (ODbL) v1.0.**

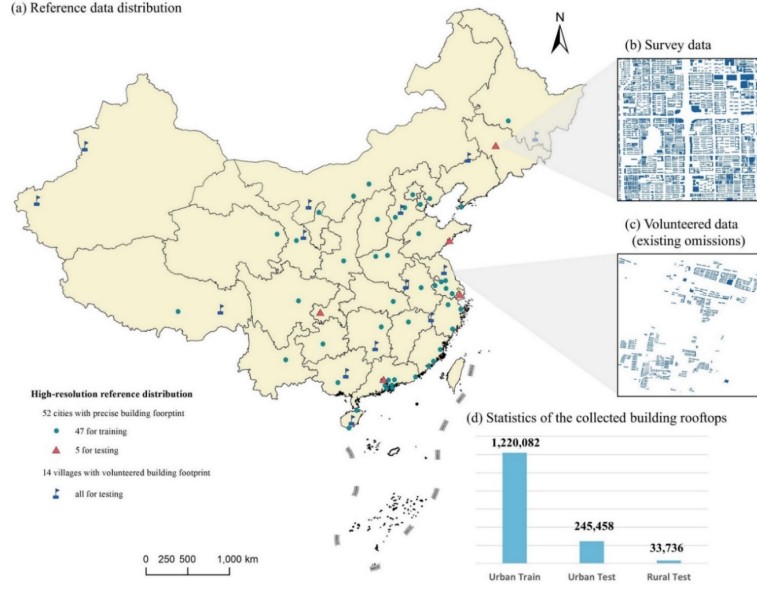

**Figure 4: Illustration of the collected high-resolution reference. (a) is the high-resolution reference distribution map (base map © OpenStreetMap contributors 2023. Distributed under the Open Data Commons Open Database License (ODbL) v1.0). (b) and (c) are real-world examples of the collected survey data (43.88°N, 125.37°E) (survey data © Tiandi-Map) and the volunteered data (33.47°N, 119.79°E) (volunteered data © OpenStreetMap contributors 2023. Distributed under the Open Data Commons Open Database License (ODbL) v1.0), respectively. (d) is the statistic of building rooftops.**



## 3.2 Reference data

For the deep-learning-based method, the supervised information (e.g., reference data) is crucial to the model performance. In this study, we collect three kinds of reference data for training and evaluation, i.e., the survey building rooftop data (2.5 m) the volunteered building rooftop data (2.5 m), and the land cover data (10 m).

The survey data should reflect the precise building rooftop distribution in the region of interest. Hence, in this study, we collect 52 cities' building rooftops for the year 2019 from Tiandi-Map, which is sponsored by the National Platform for Common Geospatial Information Service of China (Zhang et al., 2021). We use 47 cities for training (1.22 million buildings) and 5 cities for testing (250, 000 buildings), as shown in Fig. 4. To verify the accuracy in the rural area, we collect additional building rooftops of several rural regions from the volunteered geographic information platform, i.e., the Open Street Map (OSM) (Haklay and Weber, 2008). However, there are uneven omissions and errors in the OSM data. To address these issues, we manually correct the data on the ArcGis software in conjunction with high-resolution images provided by ArcGIS Online (Arcgis online, 2022). Finally, building rooftops of 14 villages are obtained (30, 000 buildings), which are free of errors despite the presence of small omissions, as shown in Fig. 4.

To improve the geospatial generalization of the deep learning method (i.e., scaling to all regions of China), we also collect the land cover data over China from 2016-2021 from Dynamic World product (Brown et al., 2022). Dynamic World, as a result of the partnership between Google and the World Resources Institute, is a near real-time (2-5 days) and 10 m global land cover dataset. It includes ten land cover types and provides the probability estimates for each type. This paper only focuses on the "built" land cover type for weakly supervised learning. Though the resolution could not meet the demand of our CBRA (2.5 m), the dynamic world could provide a kind of vital information such as "where there might be a building rooftop". The strategy of sampling Dynamic World as the training reference would be illustrated in Sect. 4.1, and how to use it as reference information for updating the parameter in our model would be clarified in Sect. 4.3.

## 4 Methodology

Fig. 5 shows an overview of the methodology workflow, including (a) the training sample generation, i.e., arranging the high-resolution reference, low-resolution reference, and Sentinel-2 imagery, (b) the proposed STSR-Seg framework for detecting the building rooftop area, which is the core of our workflow, and (c) the workflow used for BRA data generation based on the trained STSR-Seg framework. (d) The strategy for dataset evaluation.



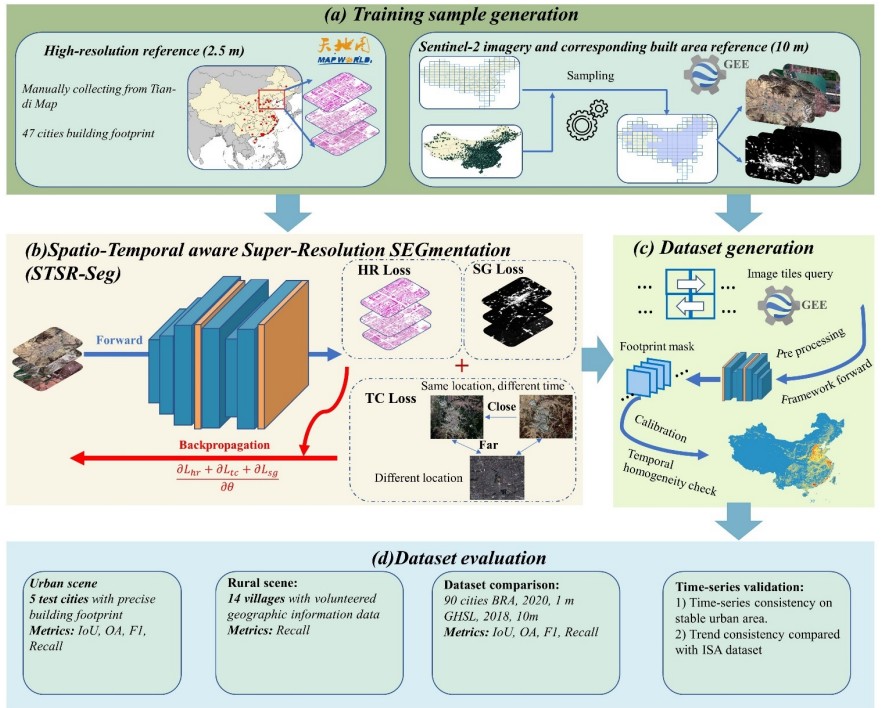

**Figure 5: The flowchart for CBRA mapping and evaluation.**

## 4.1 Training sample generation

The deep-learning-based method is data-driven, and the training samples are crucial to its generalization performance. As
described in Sect. 3, the training reference we have consists of the high-resolution building rooftop in 47 cities and the low-resolution land cover data. For the rooftop data, we can easily pair them with the Sentinel-2 imagery obtained in the same location and time. However, for two reasons, we think it is not a wise choice to use all the available land cover data for 2016-2021 in China or just uniformly sample a part from the land cover database for training. First and foremost, utilizing all the data or uniform sampling would lead to a large amount of redundancy in supervised information. For example, 13% of land in
China is deserted, and 23% is forested. The redundancy of these non-human areas would bring unbalanced categories, thus leading to ineffective model training. Secondly, China covers an area of approximately 9.6 million km², and utilizing all the land cover for training would bring a great burden on our computation recourses. To work around this, we assume that people mainly live in the vicinity of basic administrative units. We utilize the third level of China administrative divisions, i.e., the county level, for a total of 2844, as the basic units. Hence, the heuristic sampling strategy is as followings:

(1) Sampling 200 coordinates using the Gaussian distribution spanning a standard deviation of 150 *km* around each basic unit.

(2) For each coordinate, randomly assign three reference years over 2016-2021.





(3) Checking whether the coordinate exists within valid Sentinel-2 tiles for the reference year and with less than 10% cloud, then processing and downloading the image patch, as well as the corresponding land cover type. Otherwise, go to step

255    1

Through this approach, the land cover training samples, covering both urban and rural scenes and ranging for various years, are easily and automatically gathered (see results in Fig. A1). Finally, we further rebalance the gathering samples to make sure that the built type is the majority by thresholding. In all, we obtain two sets for model training. One is the high-resolution reference set, paired with the Sentinel-2 imagery (10 m) and the building rooftop reference (2.5 m). The other is the low-

resolution reference set, paired with the Sentinel-2 imagery and the corresponding land cover type (10 m). We also assign a corresponding "built" land cover type for each building rooftop reference, as high-resolution references and low-resolution references can be learned jointly in our learning strategy.

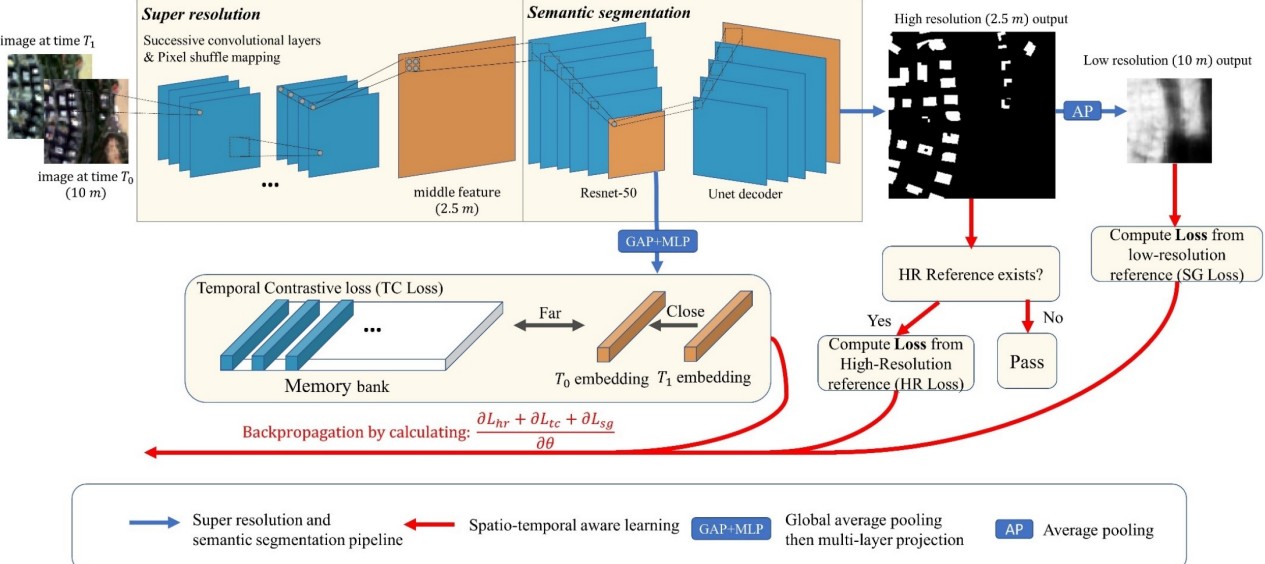

**Figure 6: The schematic diagram of the STSR-Seg framework.**

**4.2 Super resolution and semantic segmentation pipeline**

To achieve super-resolution and robust building rooftop prediction, we design the STSR-Seg framework, as shown in Fig. 5. The STSR-Seg includes two major components: the super resolution and semantic segmentation pipeline (i.e., blue forward arrow in Fig. 6), and the spatio-temporal aware learning (i.e., red backward arrow in Fig. 6). In this sub-section, we would describe the forward pipeline in details.

We utilize the SOTA method, EDSR (Lim et al., 2017), to serve as the front component of the framework (i.e., the super resolution component). Let $I \in R^{C \times H \times W}$ denote the input imagery, where $C$, $H$, $W$ are the channel, height, and width, respectively. The EDSR first utilizes successive convolutional layers embedded with residual connections to increase the dimension of $C$. For example, assuming the up-sampling factor is r, the implemented CNN would output the feature with the





dimension of $Cr^2 \times H \times W$. Then, the EDSR enlarges the $H$ and $W$ dimensions by applying the pixel shuffle function, and
outputs the fine-grained middle feature F $\in R^{C \times rH \times rW}$. In this paper, the up-sampling factor r is 4.

Next, we apply a modified Unet architecture (Ronneberger et al., 2015) to serve as the rear component (i.e., the semantic segmentation component) to obtain high-resolution and pixel-wise rooftop prediction. To enlarge the capacity of the naïve Unet, we replace the encoder of the naïve Unet with the Resnet-50 (He et al., 2016), which is a powerful and widely-used residual network. We also replace the final up-sampling layer in the decoder with a deconvolution layer, and add a batch
normalization layer in each convolutional block of the naïve Unet. With the F output by the SR, the modified Unet will predict high-resolution sigmoid confidence of the building rooftop area $\hat{P}_{high} \in R^{1 \times rH \times rW}$. To achieve a robust learning strategy, we add an additional global average pooling layer and a fully connected layer in the encoder of the Unet (i.e., Resnet-50), and output the temporal representation z $\in R^d$ of the input imagery, where $d$ is the representation dimension.  We also add an additional average pooling layer to the high-resolution prediction map $P_{high}$, and obtain the low-resolution sigmoid
confidence$\hat{P}_{low} \in R^{1 \times H \times W}$. The overall output of this SR-SS forward pipeline is of three-folds, i.e., the $\hat{P}_{high}$, the $\hat{P}_{low}$and the z. The $\hat{P}_{high}$ is what we need to generate the dataset, while the $\hat{P}_{low}$ and the z are served for producing auxiliary loss for tuning the model parameter (see in Sect. 4.3).

## 4.3 Spatio-temporal aware learning

In this paper, we regard the large-scale and multi-annual building rooftop mapping as a weakly supervised learning problem,
since the survey rooftop reference could only be gathered in a part of urban areas of a certain year as described in Sect. 3.2. To generalize both the temporal (e.g., 2016-2021) and the spatial (e.g., all over China), we design a robust model learning strategy, i.e., the spatio-temporal aware learning. Our spatio-temporal aware learning contains three learning algorithms

(1)  The High-Resolution loss (HR loss) is a fully-supervised loss, used for better learning the supervised information from the collected high-resolution survey rooftop reference.
(2)  The Temporal Contrast loss (TC loss) is a weakly-supervised loss, used to allow the model be invariant to subtle variations over time (e.g., due to image acquisition times)
(3)  The Spatial Generalization loss (SG loss) is another weakly-supervised loss, used to inform the model to generalize the spatial extent where the high-resolution survey rooftop data is not available.

### 4.3.1 High-resolution learning

The STSR-Seg gives a sigmoid confidence map of the building rooftop $\hat{P}_{high} \sim [0, 1]$ and we have the high-resolution rooftop reference $P_{high} \sim \{0, 1\}$. In our HR loss, we firstly calculate the cross entropy as follows:

$$L_{ce}\left(P_{high}, \hat{P}_{high}\right) = -\sum P_{high} log \hat{P}_{high} + \left(1 - P_{high}\right) \log\left(1 - \hat{P}_{high}\right), \tag{1}$$



Previous work on semantic segmentation has shown the mixed cross entropy loss is effective (Iglovikov et al., 2018). Here, we utilize Focal Tversky Loss (Abraham and Khan, 2019), which is a tunable loss function:

$$L_{ftl}(P_{high}, \hat{P}_{high}) = \left(1 - \frac{\sum P_{high}\hat{P}_{high} + \varepsilon}{\sum(1-\beta)P_{high} + \sum \beta\hat{P}_{high} + \varepsilon}\right)^{\gamma}, \tag{2}$$

where $\varepsilon$ is a constant providing numerical stability, $\gamma$ is the focal parameter to balance the loss weight between the easy sample ($\hat{P}_{high} \approx P_{high}$) and the hard sample. $\beta$ is the parameter to control the trade-off between the importance of false positives (FP) and false negatives (FN). In this paper, we set $\varepsilon = 10^{-6}$, $\gamma = 0.5$, $\beta = 0.6$. The $\gamma < 1$ would improve the model convergence by shifting the focus on the easy sample. Because in the informal experiment, we find that some hard samples are actually mislabelled in our training set, such focal parameter would make the model robust to the label noise. Besides, $\beta > 0.5$ would shift the convergence more on minimizing FN predictions to improve the recall score of the model. The overall HR loss is given by weighted sum:

$$L_{hr} = L_{ce} + 0.5 \cdot L_{ftl} \tag{3}$$

### 4.3.2 Temporal contrast

Sentinel-2 images at the same location but at different times have very different hues, and the model may fail to predict for images with "unseen" image styles in training samples. To tackle this, we utilize the location as a priori and encourage the temporal representation corresponding to pairs of images with the same location but different times to be semantically more similar than typical unrelated pairs (i.e., from other locations), thus making the model keep time-invariant according to the image style. This similarity could be measured by calculating matrix similarity (e.g., dot product) among the two similar representations z and $\hat{z} \in R^d$, and the unrelated representation k $\in R^d$. Here, following the previous contrastive learning framework MoCo (He et al., 2020), we implement InfoNCE as the similarity measure:

$$L_{tc}(z, \hat{z}, k_j) = -log \frac{\exp(z \cdot \hat{z}/\tau)}{\exp(z \cdot \hat{z}/\tau) + \sum_{j=1}^{N} \exp(z \cdot k_j/\tau)}, \tag{4}$$

where $\tau$ is a temperature hyperparameter scaling the distribution of the similarity measurement. For each training step, we assign the anchor image with a random selection of images from other years, and obtain the pairing temporal representation z and $\hat{z}$ from the Resnet-50. As for the unrelated representation $k$, we maintain a memory bank to store the representation from $N$ previous steps. The memory bank is a queue structure with the size of $N \cdot d$. The memory bank is first zero-initialized. For each training step, we adopt the first-in-first-out (FIFO) strategy to update the queue by adding the anchor representation z from the previous step and removing the oldest representation. In this paper, the hyperparameter $\tau = 0.75$, N = 16, and d = 128.

### 4.3.3 Spatial generalization

Though the HR loss could provide precise pixel-to-pixel supervision, this information is only available in urban regions (i.e., 47 cities) and is sorely inadequate in other regions of China, e.g., rural regions. This situation inspires us to use additional low-resolution references (e.g., LUCC data) from outside the spatial extent of our collected high-resolution survey data to better inform the model. Given the low-resolution output $\hat{P}_{low}$ and the land cover reference $P_{low}$, it is intuitive to calculate the cross-entropy loss (Eq. 1), i.e., $L_{ce}(P_{low}, \hat{P}_{low})$. Obviously, the $\hat{P}_{low}$ is an average aggregation of the $\hat{P}_{high}$, i.e., each pixel in $\hat{P}_{low}$ denotes an average $4 \times 4$ block in the $\hat{P}_{high}$ in our experimental setting. The cross entropy could suppress the prediction score of the background pixel of $\hat{P}_{low}$, i.e., non-building pixel, and also suppress the corresponding $4 \times 4$ pixels in $\hat{P}_{high}$. However, for the foreground pixels, the cross entropy homogeneously boosts the prediction score for all pixels related to the built area in $\hat{P}_{low}$, introducing errors to $\hat{P}_{high}$ like false predictions to roads and city squares. To tackle it, the loss must be interpreted in a softer manner, which means the prediction score should not be uniformly improved.

In the case of BRA mapping, the descriptions of "built" class in Dynamic World product (10 m) suggest it is a mixture of building and other impervious surfaces (Table B2). Therefore, we assume each low-resolution built land cover determines a known distribution over frequencies of the high-resolution building rooftop (Fig. A2). Inspired by the success of super-resolution loss (Malkin et al., 2018), we utilize a variant of it, which encourages our model to match its $\hat{P}_{high}$ to the fixed distributions obtained by the low-resolution reference. Specifically, we assume the high-resolution building rooftop $c_{hr}$ follows the Gaussian distribution in the low-resolution built-up area $c_{lr}$, i.e.,

$$P_{low}(c_{hr}|c_{lr}) = \mathcal{N}(\mu, \sigma^2), \tag{5}$$

where $\mu$ and $\sigma$ are the mean and standard deviation of the reference Gaussian distribution, respectively. They could be statistically obtained from our training set where both the high-resolution and low-resolution references are available, or be manually set. Also, due to the $\hat{P}_{low}$ is derived from $\hat{P}_{high}$ by averaging, the $\hat{P}_{low}$ also follows an estimated Gaussian distribution $\mathcal{N}(\hat{\mu}, \hat{\sigma}^2)$. Therefore, the loss could be interpreted by the KL divergence of these two distributions. This optimization criterion is softer due to the statistics matching rather than the distribution fitting (e.g., the cross-entropy). Finally, we incorporate this metric into the cross-entropy loss function, and our SG loss is formulated by:

$$L_{sg}(P_{low}, \hat{P}_{low}, \mu, \sigma, \hat{\mu}, \hat{\sigma}) = L_{ce}(P_{low}, \hat{P}_{low}) + D\_KL(\mu, \sigma, \hat{\mu}, \hat{\sigma}), \tag{6}$$

where $\mu = 0.44$, $\sigma = 0.01$ based on the statistic of the high-resolution and low-resolution reference pairs (Fig. A2). The SG loss only utilizes low-resolution references, and could be implemented on collected land cover samples covering multiple geographies and years, thus improving the capacity of generalizing the vast geospatial mapping.

To sum up, the spatio-temporal aware learning includes three objective functions: (1) The HR loss, providing pixel-to-pixel high-resolution supervision. (2) The TC loss, learning invariance to image differences due to different times. (3) The SG loss, learning weak information from land cover samples. In the training phase, these losses are weighted to update the model parameters simultaneously:





$$L = \alpha L_{hr} + \varphi L_{tc} + \omega L_{sg}, \tag{7}$$

In our offline experiment, we have found a ratio of $\alpha : \varphi : \omega = 200 : 1 : 5$ balances the three losses effectively in our experiment. The backpropagation pipeline is illustrated in Fig. 6 (red backward arrow).

**4.4 CBRA Dataset generation**

We first download Sentinel-2 imagery covering China from 2016-2021 (Fig. 3) with a fixed grid of $0.10° \times 0.10°$. To avoid the uneven transition or stitched problem between the splicing gap of the prediction result of cropped smaller images, the rooftop is predicted by the trained model in an expansion style, which consists of five steps as shown in Fig. 7: (1) Expanding the size of the downloaded image to contain an integral number of sliding windows that overlap each other by zero padding.
(2) An $H \times W$ sliding window is created to extract image patches. During the movement, the window would assure that the next move overlaps the previous by 10% pixels. Then, the image is cropped into smaller image patches with a size of $H \times W$. (3) The cropped images are input to the model, and the sigmoid confidences of the building rooftop are obtained. (4) Calculating the maximum value of overlapping area at each pixel, then stitching the confidence map into one and removing the zero padding. (5) Utilizing 0.5 as the threshold value to distinguish between the candidate foreground pixels (i.e., building
rooftop) and the background pixels.

For the binary mask obtained from the expanding prediction, the intersection is then taken between our prediction and the built area provided by Dynamic World, to remove any candidate pixels that do not intersect with the built area. This process would reduce the false positives because our model potentially incorrectly identifies the bare land as the building rooftop. The built area provided by Dynamic World is a possibility estimation ranging from 0-1. We utilize a low threshold of 0.2 to
distinguish between built-up and unbuilt-up areas.

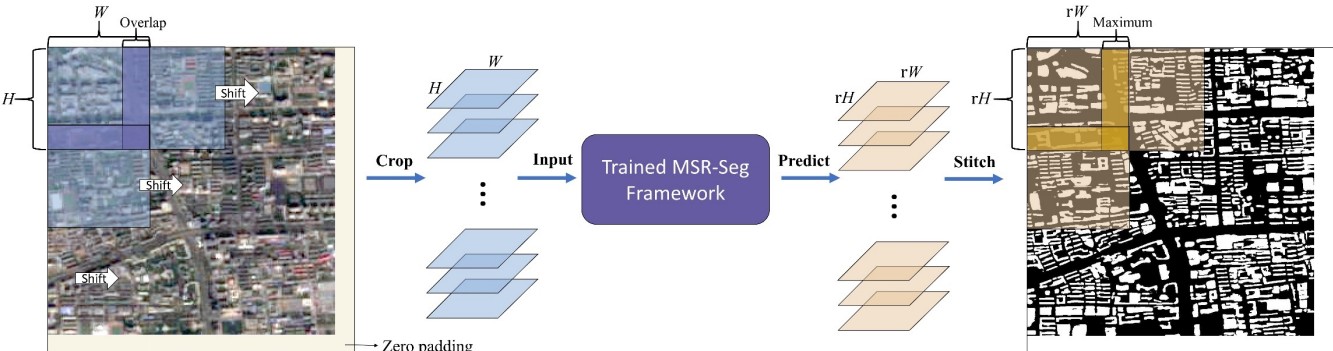

**Figure 7: Schematic diagram of our dataset generation workflow. Imagery (left) © ESA.**

As previous research (Radoux et al., 2014) demonstrates the transition pixels between two different land-cover types are potentially misclassified, the temporal homogeneity of each candidate pixels is checked inspired by Li et al, (2015), as shown
in Fig. 8. For each pixel, a sliding $3 \times 3$ window is employed to determine the final pixel value by majority voting. This




ensures that the results are similar in the adjacent years. However, for the edge year like 2016 and 2021, they are not checked due to the lack of temporal information.

The implementation is conducted on our local server with $2 \times$ NVIDIA Tesla P40 GPU. The overall dataset generation pipeline costs about 3 months using Python.

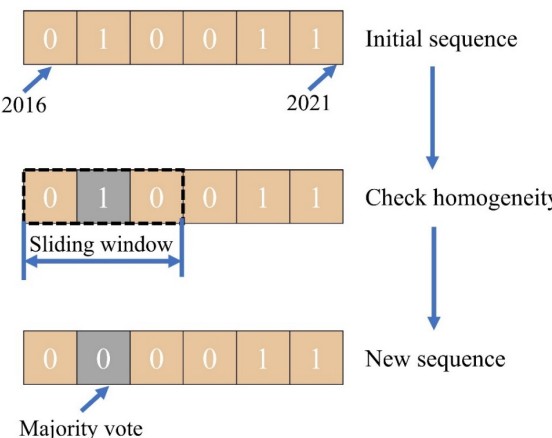


**Figure 8: The temporal homogeneity check.**

**4.5 Accuracy assessment**

To comprehensively assess the performance of our 2.5 m multi-annual CBRA dataset. The sampled-based approaches and temporal-based approaches are adopted. Firstly, the sample-based approaches utilize five cities (250, 000 buildings) with

precise building rooftops for testing (Fig. 4). The metrics are listed in Table 2. Accuracy, intersection-over-union (IoU), recall, and F1-score all range from 0-1, and 1 indicates the best classifier. In rural areas, however, there is a lack of reliable high-resolution references. As described in Sect. 3.2, we utilize our manually-calibrated OSM data as the reference. There are 14 villages in all (Fig. 4), accounting for 30, 000 buildings. Since there are still a few omissions, we only examine the recall in rural areas. Buildings are dynamic and may change each year (van Etten et al., 2021). To ensure the reliability of the evaluation

accuracy, we only use the prediction results of the corresponding year for accuracy evaluation, e.g., 5 cities correspond to 2019 and 14 villages correspond to 2020. As for the dataset comparison, we use two products for comparison: (1) China 90 cities 1 m building rooftop area dataset (90-cities-BRA) (Zhang et al., 2022b). (2) The 10 m global human settlement layer (GHSL) (Corbane et al., 2021). To our best knowledge, the 90-cities-BRA is currently the only large-scale and freely accessible building rooftop data in China, and it covers 90 cities for 2020, which is also the dataset we can compare in the urban scene at a fair.

The GHSL provides the human settlement layer of the globe for 2018, and we mainly use it for the comparison of rural scenes.

Secondly, in the temporal-based approaches, we design two experiments to estimate our performance consistency in a time span. For the first experiment, we assume that the buildings in the old town area of Beijing, Hong Kong and Macao remain unchanged in the last six years, therefore, we test the consistency of the results in terms of evaluation metrics in these regions. For the second experiment, we calculate the correlation coefficient (e.g., $R^2$) of our data with the existing well-known annual



impervious surface products. To achieve it, we utilize 30 m China land cover dataset (CLCD) (Yang and Huang, 2021) and 30 m global artificial impervious area (GAIA) (Gong et al., 2020b), ranging from 1990 to 2019 and 1985 to 2018, respectively.

As well as evaluating our data, we examine several examples of our poor rooftop extraction result, to understand the limitation of our dataset.

**Table 2: Classification performance metrics calculated in this study.**

| Metric | Definition |
|---|---|
| True positive (TP) | Pixels correctly classified as positive (i.e., building rooftop). |
| False positive (FP) | Pixels incorrectly classified as positive. |
| True negative (TN) | Pixels correctly classified as negative (i.e., background). |
| False negative (FN) | Pixels incorrectly classified as negative. |
| Intersection over union (IoU) | TP/(TP+FP+FN). |
| Recall | TP/(TP+FN) |
| F1-score | 2×TP/(2×TP+FP+FN) |
| Overall accuracy (OA) | (TP+TN)/(TP+FP+TN+FN) |

**5 Results**

The implementation configurations of the overall generation pipeline are listed in Table B3. Based on all available Sentinel-2 data in China, we generate the annual 2.5 m resolution CBRA (China-building-rooftop-area) dataset of 2016-2021. To evaluate it, we first use independent testing samples to assess the performance of CBRA in urban and rural areas and compare it with other datasets, both qualitatively and quantitatively (Sect. 5.1). Then, we test the time consistency of the CBRA by using stable

samples and other ISA datasets (Sect. 5.2). Finally, we analyse the BRA in China of 2016-2021 in terms of spatial distribution and temporal change (Sect. 5.3).

**5.1 Accuracy assessment using testing samples**

**5.1.1 Quantitative analysis**

The accuracy of CBRA is first assessed via the collected samples from urban scenes and rural scenes. The confusion matrix

for building rooftop identification in urban scenes is given in Table 3, and the performance statistics in both urban and rural areas are given in Table 4.

In urban scenes, although the CBRA is derived from Sentinel-2 imagery (10 m), it achieves a balance result in TP and TN with a higher F1-score value of 62.55% (+ 10.61%) compared to the previous 90-cities-BRA, which is derived from high-resolution GES imagery (1 m). In terms of IoU, the CBRA obtains a score of 45.51%, indicating the CBRA has a high

classification accuracy for building rooftop pixels. In addition, the OA is slightly lower than 90-cities-BRA (-0.54%), this is due to the several blob-like predictions of the CBRA because of relatively low resolution (2.5 m) compared with 90-cities-BRA (1 m), which will be covered in more detail in Sect. 6.2. For recall, the CBRA obtains 74.66%, which achieves great improvement (+ 27.29%) compared with 90-cities-BRA, mainly because our robust designation of STSR-Seg framework.



In the rural scene, there is no publicly available building rooftop dataset in rural areas of China before our CBRA, hence we
compare the CBRA with GHSL, which is a human settlement layer data (resolution of 10 m), and we only evaluate them in
terms of recall. The GHSL is a result of a coarser level compared with the building rooftop (e.g., including impervious surfaces
like roads and city squares), thus achieving the highest recall value (80.89%) in rural scenes. However, the CBRA is very close
to it (78.94%), with a gap of only 1.95%, indicating a reliable prediction in rural areas.

**Table 3:** Statistic of the confusion matrix for building rooftop extraction in urban scenes.

| Dataset | TP (%) | FP (%) | TN (%) | FN (%) |
|---|---|---|---|---|
| 90-cities-BRA (Zhang et al., 2022b) | 14.32 | 12.29 | 68.52 | 4.86 |
| CBRA (Ours) | 8.98 | 6.65 | 74.42 | 9.96 |


**Table 4:** Performance metrics for building rooftop extraction*.

| Dataset | Description | Urban scenes | | | | Rural scenes |
|---|---|---|---|---|---|---|
| | | IoU (%) | OA (%) | Recall (%) | F1-score (%) | Recall (%) |
| 90-cities-BRA (Zhang et al., 2022b) | 90 cities building rooftop in China with a resolution of 1 m (2020) | 35.08 | 83.39 | 47.39 | 51.94 | - |
| GHSL (Corbane et al., 2021) | Global human settlement layer with a resolution of 10 m (2018) | 25.85 | 53.84 | 84.94 | 41.07 | 80.89 |
| CBRA (Ours) | China building rooftop data with a resolution of 2.5 m (2016-2021) | 45.51 | 82.85 | 74.66 | 62.55 | 78.94 |

* 90-cities-BRA does not include the rural area in China.

**5.1.2 Qualitative analysis**

To further test the performance of the CBRA, we select several examples from our testing set to analyze and compare our
results in both urban and rural areas. As shown in Fig. 9, in urban areas, our CBRA and the 90-cities-BRA are generally similar
in the region where buildings are well separated, e.g., Fig.9a2, and Fig. 9b2. The difference is mainly about the rooftop details,
the CBRA ignores several vertexes on the boundary, thus achieving blob-like results, which is mainly due to the resolution
gap, as shown in Fig. 9c1 and Fig. 9d1. In the dense urban areas, especially in the old town where the distance between
buildings may be less than 2.5 m, the CBRA treats buildings as blocks, e.g., Fig. 9a1. However, the CBRA has more complete
building rooftops and fewer false predictions on the background (e.g., the road) compared with 90-cities-BRA, as shown in
Fig. 9c2 and d2, which explains the smaller value of FP of CBRA in Table 3. Besides, the 90-cities-BRA utilizes the Google
Earth Satellite (GES) images as the data source. Although GES images are of high spatial resolution (e.g., 1 m), GES images
are provided by different satellites simultaneously and do not have consistent geographic offsets and acquisition times. The
CBRA utilizes a super resolution technique to extract 2.5 m results only from the Sentinel-2 satellite, ensuring the reliability
of the geography and the acquisition time, as shown in Fig. 10 and 11.





In rural areas, as shown in Fig. 12, CBRA also provides building rooftop areas, while the 90-cities-BRA does not include them. Although it is difficult to identify individual buildings from the Sentinel-2 images, CBRA still extracts them, as shown in Fig. 12e and f. Compared to other datasets that provide information related to buildings in rural areas, the CBRA is at a significantly fine-grained scale (Fig. 13).

In summary, the CBRA achieves higher performance on extracting building rooftop (TP) and suppressing the false prediction on the background (FP), with 62.55% (+ 10.61%) in term of F1-score compared with the 90-cities-BRA. Besides, the CBRA has a full coverage of China, including the rural areas, which is at a finer scale than other existing full-coverage and thematic-related products. In addition, the temporal coverage of CBRA spans 6 years (2016-2021), which is the first available building rooftop data with a span of time. The temporal information in the CBRA would be analysed in the subsection

465    5.2.

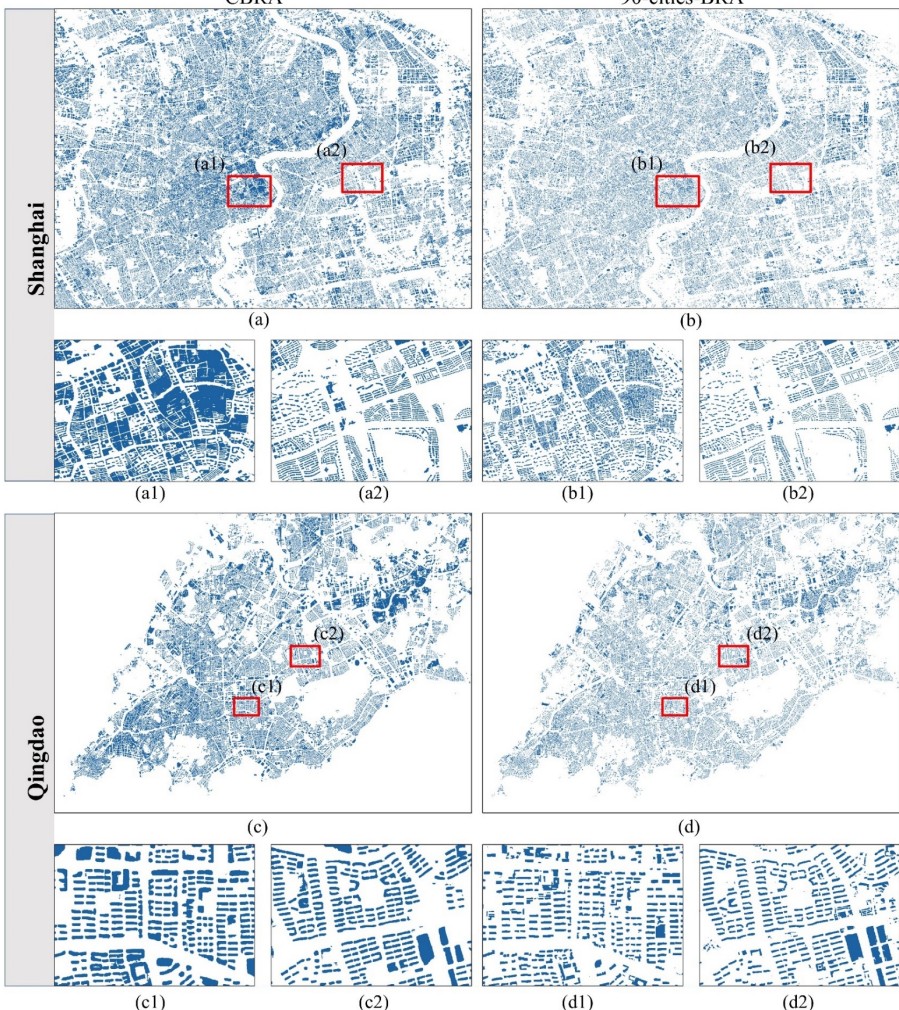

**Figure 9: Comparison of the CBRA and the other dataset over the sampled urban regions in Shanghai and Qingdao. a and c are results of our CBRA. b and d are results of the 90-cities-BRA (Zhang et al., 2022b).**



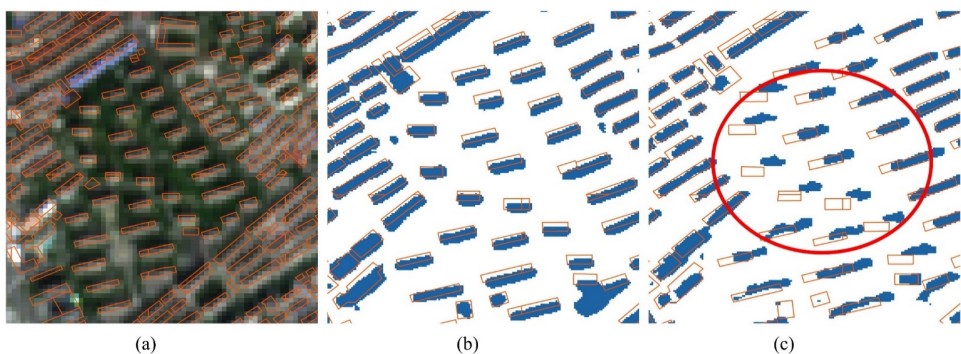

**Figure 10: Example of the inconsistent geographical offset of the previous dataset (121.531467° N, 31.299903° E). (a) The Sentinel-2 image with survey rooftop data (imagery © ESA). (b) Result of CBRA. (c) Result of 90-cities-BRA (Zhang et al., 2022b). On could observe that the result from 90-cities-BRA has geographical offset as red circle indicates. The CBRA uses the imagery only from Sentinel-2 satellite, ensuring the reliability of the geographical positions.**

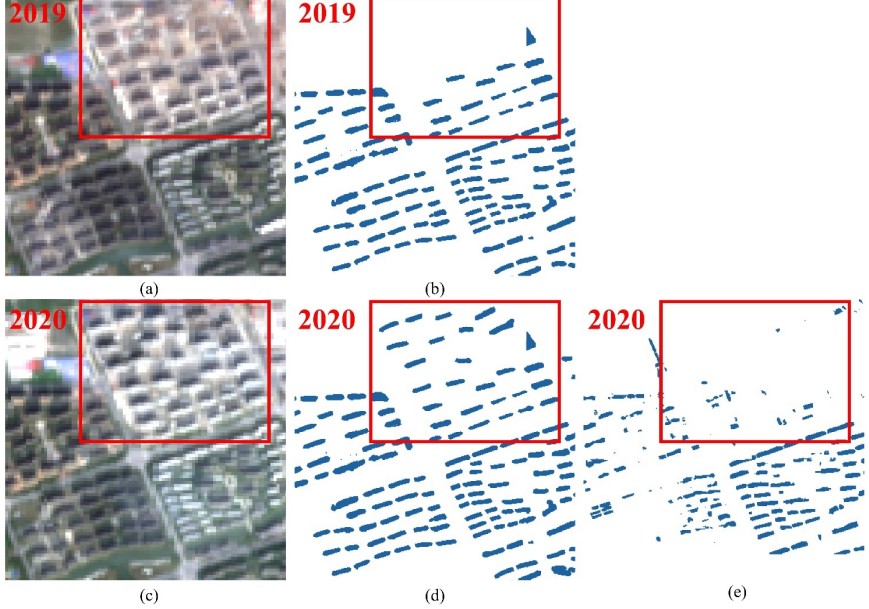

**Figure 11: Example of the inconsistent acquisition time of the previous dataset (121.341982° N, 30.762489° E). (a) The Sentinel-2 image in 2019 (imagery © ESA). (b) Result of CBRA in 2019. (c) The Sentinel-2 image in 2020 (imagery © ESA). (d) Result of CBRA in 2020. (e) Result of 90-cities-BRA (Zhang et al., 2022b) in 2020. The CBRA uses the image with specific acquisition time, ensuring the reliability of the results in terms of temporal consistency.**

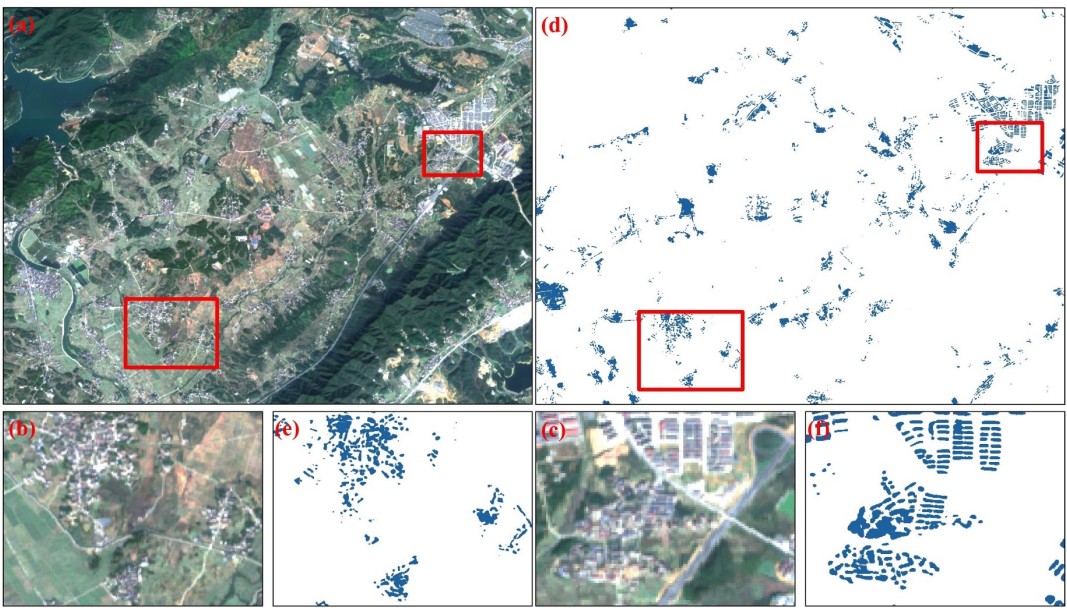

**Figure 12: Example of the rural area (118.328041° N, 28.817881° E). (a-c) Sentinel-2 images (imagery © ESA). (d-f) Results of CBRA.**

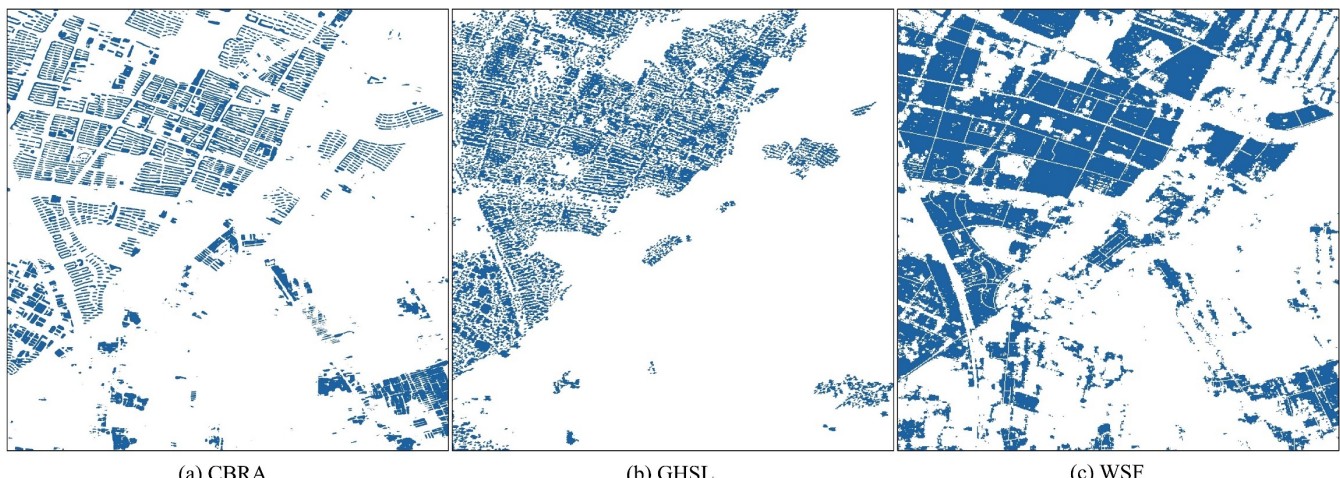

| (a) CBRA | (b) GHSL | (c) WSF |

**Figure 13: Comparison of the CBRA and the other datasets over the sampled rural region (106.352257° N, 38.533718° E). (a) CBRA. (b) GHSL (Corbane et al., 2021). (c) WSF (Marconcini et al., 2020b).**

## 5.2 Temporal consistency analysis

To evaluate the temporal characteristics of the CBRA, we first test the performance of CBRA in three regions, i.e., the old town of Beijing, Hong Kong and Macao, where the distribution of the building is almost stable without change based on our priori knowledge. We utilize the survey rooftop data collected in 2019 to quantitatively demonstrate the accuracy, as shown in Fig. 14. Overall, the CBRA shows good performance consistency with little variation over 2016-2021. One may observe the accuracy fluctuates between 2016 and 2017, there are two potential reasons for this. First is the relatively long interval

between the sampling time of survey data (2019) and the year 2016. The second is that the results for 2016 are not checked by temporal homogeneity due to the lack of temporal information; thus, its reliability is slightly lower compared to other years.

The well-known annual impervious surface area (ISA) products could provide time span information for the evaluation. Thereby, we compare CBRA with the ISA of CLCD (CLCD-ISA) (Yang and Huang, 2021) and the GAIA (Gong et al., 2020b). We calculate fractions of foreground pixels within the $0.10° \times 0.10°$ spatial grid for each year and then estimate the

correlation coefficients ($R^2$) between CLCD-ISA and GAIA to quantitatively demonstrate their agreement. Overall, the CBRA shows good consistency with the ISA products over the time span ($0.63 < R^2 < 0.71$), indicating the reliability of the CBRA (Fig. 15).

Although good agreement has been found between 2016-2019, the 2020 and 2021 are not checked because the annual ISA products with close resolution are not available for these years. However, the training material for producing CBRA contains

the Dynamic World (Brown et al., 2022), which is a timely updated product providing built land cover, and the CBRA is therefore in very good agreement with it from 2016 to 2021 ($0.83 < R^2 < 0.89$), also indicating the reliability of the CBRA.

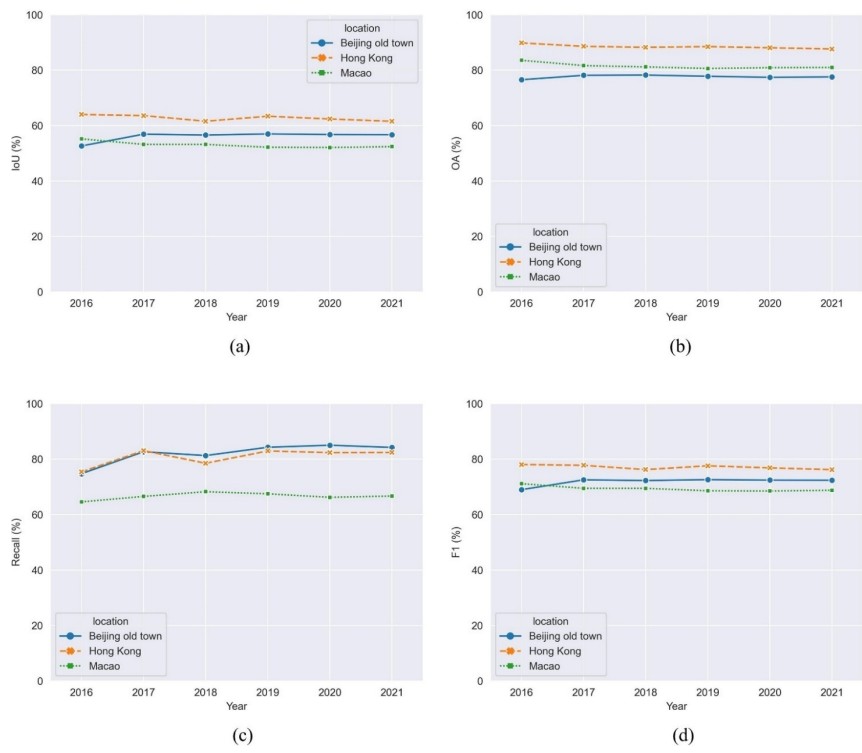

**Figure 14: Accuracy of CBRA in the building stable regions over 2016-2021. (a) IoU. (b) OA. (c) Recall. (d) F1-score.**



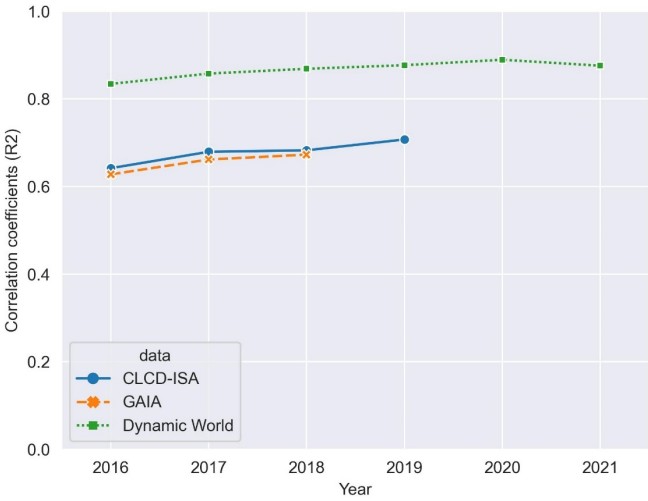


**Figure 15: The correlation coefficients of the fraction of the foreground pixels between CBRA and two thematic-related datasets for each year. The fraction is aggregated within the $0.10° \times 0.10°$ spatial grid.**

**5.3 The spatial and temporal characteristics of China BRA from 2016-2021**

The statistical result of the average area of building rooftops in China from 2016-2021 is shown in Fig. 16. From the perspective

of spatial distribution, there are three main city clusters in China: (a) The North China Plain (NCP), (b) the Yangtze River Delta (YRD); and (3) the Greater Bay Area (GBA). The NCP is the largest alluvial plain in China. There are 19.8% of the population living here (280 million out of 1.4 billion), but occupying 30.4% building rooftop areas (27,277 out of 89,826 $Km^2$), which indicates a more developed primary and secondary industry in the region (more industrial buildings and farm buildings). The YRD is dominated by Shanghai and is one of the regions with the most active economic development, providing 24.1%

of GDP of China. There are 16.4% population (236 million) living here, and occupying 16.0% building rooftop area (14,342 $Km^2$). The ratio of population and building areas are almost equal, indicating a more developed tertiary industry in the region. The GBA is a city cluster consisting of 11 cities including Guangzhou, Shenzhen, Hong Kong and Macao, which is the largest and most populated urban area in the world. There are 6.0% population (86 million) but occupying only 3.9% building rooftop areas (3,472 $Km^2$), indicating the region has a developed tertiary industry along with a high density of population and a tighter

housing supply.

Fig. 17 quantitatively summarizes the BRA and their changes on the three city clusters from 2016 to 2021. Overall, the China BRA has increased over the past 6 years, with more than 110,000 $Km^2$ in 2021, which is increased by 34,000 $Km^2$ compared with 2016 (Fig. 17a). In addition, Fig. 17b indicates that the proportion of BRA on the NCP and YRD has obviously increased, while the proportion of the GBA and other regions except these three city clusters has slowly declined from 2016-

2021. Specifically, the proportion of the NCP increases the most, from 27% to 31%, while the proportion of other regions clearly decreases, from 53% to 49%. The change in the proportion reveals that the urbanization in China is characterized by

the concentration of large city clusters. Lastly, Fig. 17c and d illustrates the statistic of BRA change from 2016 to 2021 and the expansion area on each city clusters, respectively. Specifically, the NCP has a largest increased with a total of 13,081 $Km^2$ (from 20, 884 $Km^2$ in 2016 to 33, 966 $Km^2$ in 2021).

The spatial distribution of the temporal changes of building rooftop area in China is shown in Fig. 18. It could be observed that the BRA in developed regions, such as coastal regions, is increasing, while the BRA in less developed regions, such as the northeast, northwest and southwest regions of China, is decreasing.  Fig. 18(b) and Fig. 18(c) are two examples of building demolition and construction, showing the removal of dense buildings (e.g., shack houses) in the rural area, and the establishment of buildings (e.g., apartments) in the urban area, respectively. For simplicity, we only show building dynamic

in a one-way conversion pattern.

Overall, we establish the relationship between the BRA of China with the natural and economic spatial difference, which also validates the accuracy of the CBRA. The analysis on its temporal change reveals the spatio-temporal trends of BRA in China. Further analysis will be left for exploration in the future.

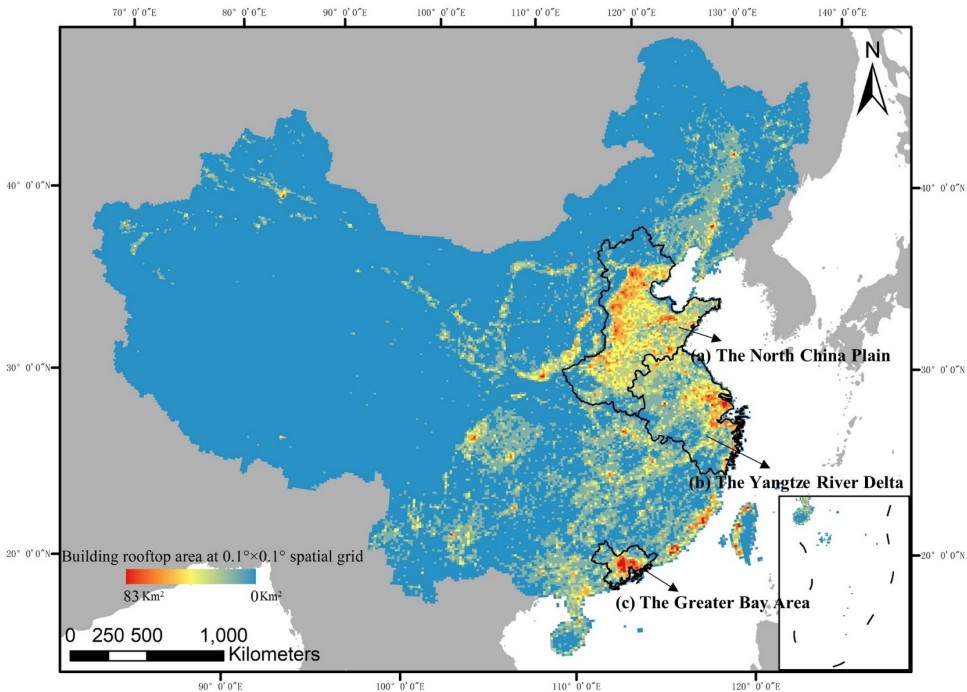

**Figure 16: The spatial distribution of the average area of building rooftops in China over the period of 2016-2021. The area fraction is aggregated within the 0.10° × 0.10° spatial grid. Base map © OpenStreetMap contributors 2023. Distributed under the Open Data Commons Open Database License (ODbL) v1.0.**



**Figure 17: The change of building rooftop area of China and three biggest city clusters in China (NCP, YRD and GBA) over the period of 2016-2021. (a) The annual statistic of building rooftop area in China. (b) The proportion of building rooftop of the biggest city clusters in China and other regions from 2016 to 2021. (c) The annual statistic of building rooftop area of three biggest city clusters and other regions. (d) The increased building rooftop area on each city clusters and other regions.**



**Figure 18: (a) The spatial distribution of the annual change of building rooftop area over the period of 2016-2021. The area fraction**
**is aggregated within the $0.10° \times 0.10°$ spatial grid (base map © OpenStreetMap contributors 2023. Distributed under the Open**
**Data Commons Open Database License (ODbL) v1.0). (b) An example of the demolition of the building (113.130952° N, 22.948144°**
**E) from 2016 to 2021. (c) An example of the construction of the building (116.275761° N, 39.844715° E) from 2016 to 2021.**

# 6 Discussion

## 6.1 Importance of the spatio-temporal aware learning

In this paper, we develop a deep learning framework STSR-Seg for robust building rooftop extraction. The overall framework
contains a super resolution pipeline for up-sampling the input resolution, a semantic segmentation pipeline for obtaining pixel-
wise building rooftop classifications, and the designed spatio-temporal aware learning with three dedicated learning algorithms
(i.e., loss functions). Here, we mainly ablate the three dedicated designed learning algorithms to reveal their importance.

The baseline is the naive structure of super-resolution and semantic segmentation pipelines, i.e., EDSR as the super-
resolution module, our modified Unet as the semantic segmentation head, and the loss function is only the binary cross entropy



loss (Eq. 1). The results of our ablation experiments are shown in Fig. A3. By disabling each learning strategy in turn from the baseline, we observe the impact on F1 testing performance: SG has the most significant effect, followed by the TC.

The SG loss (Eq. 6) is designed to leverage the information from full-coverage, but low-resolution land-cover data, to achieve larger-scale weak supervision for the model training. Essentially, to achieve the SG loss, one needs to increase the amount of training resources, and therefore greatly improves the accuracy of our data-driven method (+2.38% in terms of F1 score). In addition, we qualitatively find that using SG loss will prevent the model from falling into unexplained repeated predictions, as shown in Fig. A4. Without utilizing SG loss as supervision, the model could only converge to a limited number of training resources, i.e., the collected data. When applying to a large scale (e.g., national scale), the complexity of background in remote sensing images would significantly increase, which causes serious false alarms due to larger intra-class variance, therefore resulting in the unexpected false predictions in Fig. A4. Utilizing SG loss could suppress such false alarm by providing accurate non-building supervisions.

The TC loss (Eq. 4) is proposed to keep the model time-invariant, which is essential for generating the multi-annual dataset. As shown in Fig. A5-A7, utilizing TC loss would increase the model capacity of handling time information, especially for supressing the accuracy gap between the year 2016 and 2017. Among these evaluation metrics, utilizing TC loss brings a greater improvement on recall (Fig. A5c, Fig. A6c, Fig. A7c), which indicates the TC loss would decrease the omissions of the rooftop prediction due to the different image styles of different years, thus improving the robustness of the model. Meanwhile, utilizing TC loss increases the overall performance of our method compared with the baseline (+1.46% in F1 score).

The HR loss (Eq. 3) is composited of two losses, i.e., the cross entropy loss (Eq. 1), and the Focal Tversky Loss (Eq. 2). Here we only ablate the Focal Tversky Loss in our "+HR loss" setting. Utilizing the Focal Tversky Loss would bring 0.45% improvement in terms of F1 score by shifting the model convergence more on minimizing FN predictions, and further supresses the false predictions on the background.

As a conclusion of the ablation study, the designed learning strategy in the STSR-Seg framework leads to three significant benefits: (1) The SG loss provides enough supervision all over the China, thus increasing the geographical robustness of the model. (2) The TC loss keeps the model invariant to time span, increasing the temporal robustness of the model. (3) The HR loss is an optimized loss of the naïve cross entropy loss by introducing the Focal Tversky Loss. It could slightly improve the overall performance of the model. These advantages are also complementary to each other without conflict when used jointly together.

## 6.2 Limitations and prospects

As clear from this study, our STSR-Seg framework is scalable, allowing larger areas to be monitored (e.g., national scale). However, there are still some limitations. Firstly, the building rooftops can only be identified as a single one only if the distance between two rooftops is large than 2.5 m. Some rooftops in the old town of the city (Fig. 9a), and rooftops in some villages where the buildings are close to each other could only be drawn as a block (Fig. 12f) in the CBRA. Besides, the CBRA needs



to be further improved in terms of the delineation of the building boundaries. Buildings differ from other objects of interest in that they have regularized boundaries (e.g., polygons made of lines and vertexes), and we use a dense pixel-to-pixel classification method that does not consider the morphology of the building, which makes our result a blob-like shape. For example, in Table 5, we add a buffer with 1-2 px to the collected reference rooftop data and then use this as a benchmark to calculate the accuracy. It is noted that the there is a significant increase in the TP percent (+ 4.35% for 1 px and + 6.18% for 2 px), and by a greater percentage than the increase of the FN (the increase of FN is due to the excessive background pixels considered as the ground truth). This indicates that the CBRA results suffer from ambiguous localization on the building boundaries, with an offset of 1-2 pixels from the ground truth.

We have noticed that there are many studies on the morphology extraction of buildings in recent years, such as instance segmentation methods (Liu et al., 2022; Zhu et al., 2021; Huang et al., 2021a). We also try to replace our semantic segmentation branch with current instance segmentation methods, e.g., recurrent neural network methods (Liu et al., 2022). However, the results are not good and even fail, mainly because these methods are designed for very high-resolution aerial images (sub-metric level). In addition, the efficiency of these methods is too low to support national-level building mapping.

Many endeavors utilize a post-processing strategy, e.g., Douglas–Peucker algorithm, to regularize the building boundary (Wei et al., 2019; Chen et al., 2020; Zorzi et al., 2021) and such strategy has shown the success in building mapping at a relatively small scale (Wei et al., 2019). However, in the CBRA, the use of post-processing would introduce errors due to several block estimations in the rural area and the old town of the city as aforementioned. Considering the reliability of the results, we do not post-process the boundary of the rooftop in the CBRA. But we have developed the code (shown in Sect. 8) for post-processing, and users can selectively consider such regularization or not according to their study area when using the CBRA.

The CBRA provides full-coverage and multi-annual information of building rooftops for China at 2.5 m spatial resolution, and the proposed STSR-Seg offers an opportunity to obtain high-resolution output by using relative low-resolution remote sensing images. In the future, we will work on two aspects. First, we would consider using various datasets as training resources (e.g., the CBRA and other building data at a large scale) to achieve a larger and more robust building rooftop extractions. Besides, we would focus on investigating new method for more accurate building boundary extractions.

**Table 5:** Statistic of confusion matrix for building rooftop extraction in urban scene. The collected reference is added by the buffer zone on the boundary with 1px and 2px, respectively.

| Buffer size | TP (%) | FP (%) | TN (%) | FN (%) |
|---|---|---|---|---|
| + 0px | 14.32 | 12.29 | 68.52 | 4.86 |
| + 1px | 18.67 (+ 4.35) | 7.95 | 64.95 | 8.43 (+ 3.57) |
| + 2px | 20.50 (+ 6.18) | 6.12 | 62.42 | 10.96 (+ 6.10) |



## 7 Conclusion

In this study, we propose a robust Spatio-Temporal aware Super-Resolution SEGmentation (STSR-Seg) framework for fine-grained spatial information extraction of BRA from the abundant availability of low-resolution imagery. Specifically, the
STSR-Seg framework is built on the super resolution and semantic segmentation pipeline. Given the input low-resolution image, the STSR-Seg first extracts the corresponding high-resolution feature and then achieves pixel-to-pixel classification by the semantic segmentation branch. Considering the lack of reliable building rooftop references in China, we designed the spatio-temporal aware learning to enable the model to generalize in both large geographical regions and long time period. Ablation experiments on the designed learning strategy show the complementary advantage on handling false positives of the
complex background and the temporal consistency over a time span, and the improvement of 4.29% in terms of F1 score compared to our baseline method.

The resulting China building rooftop area dataset (CBRA) is the first multi-annual (2016-2021) and full-coverage BRA dataset in China, with 2.5 m spatial resolution. The OA and F1 scores of CBRA exceed 82% and 62% respectively based on the independent testing samples in urban areas. The inter-comparison between the CBRA and the previous 90-cities-BRA
(Zhang et al., 2022b) confirms the superiority of the results obtained in this study. In particular, for the first time, the BRA in rural areas of China is further identified at a fine-grained scale compared with other building-related products. Based on the CBRA and other annual ISA datasets, the building rooftop dynamic over a time-span is also evaluated and discussed. The CBRA completes the BRA in China and will allow for a more comprehensive characterization of climate change, urban planning, and policy decisions combined with other data, such as BRA provided by Google and Microsoft. The proposed
STSR-Seg framework could also be applied for large-scale and dynamic high-resolution BRA monitoring without any data expenditure. In the future, we plan to investigate improving the BRA accuracy and extend the spatial coverage to reveal the global BRA dynamic at the 2.5 m resolution.

## 8 Data availability

The source code of the STSR-Seg and the dataset generation pipeline could be found on our project page:
https://github.com/zpl99/STSR-Seg. The 2.5 m multi-annual China building rooftop area (CBRA) dataset from 2016 to 2021 is free to access at: https://doi.org/10.5281/zenodo.7500612 (Liu et al., 2023). The CBRA is organized as GeoTIFF (.tif) raster file format with a single band and GCS_WGS_1984 coordinate system. The pixel values are 0 and 255, with 0 representing the background and 255 representing the building rooftop area. Furthermore, to facilitate the use of the data, the CBRA is split into 215 tiles of 2.5° × 2.5° spatial grid, named "CBRA_year_E/W**N/S**.tif", where "year" is the sampling year, the
"E/W**N/S**" is the latitude and longitude coordinates found in the upper left corner of the tile data.





*Author contributions.* HT conceived the study. ZL performed the investigation. HT and ZL designed the methodology. ZL developed the software. ZL, LF, SL performed the validation. ZL prepared the original draft of the paper and HT reviewed it.


*Competing interests.* The authors declare that they have no conflict of interest.

*Acknowledgment.* The authors greatly appreciate the free access to the Sentinel data provided by ESA, the Dynamic World product provided by Google, the survey rooftop data provided by Tiandi-Map, and the 90 cities building rooftop area data
provided by Nanjing Normal University. We would also like to thank the Google Earth Engine team for their excellent work to maintain the planetary-scale geospatial cloud platform, as well as the Geemap python package for interactive mapping with Google Earth Engine, developed by Qiusheng Wu.

*Financial support.* This research has been supported by the National Natural Science Foundation of China under Grant No.
42192584 and 41971280, and by the Key Laboratory of Environmental Change and Natural Disaster of the Ministry of Education, Beijing Normal University (Project No.2022-KF-07).

**Appendix A: Figures**

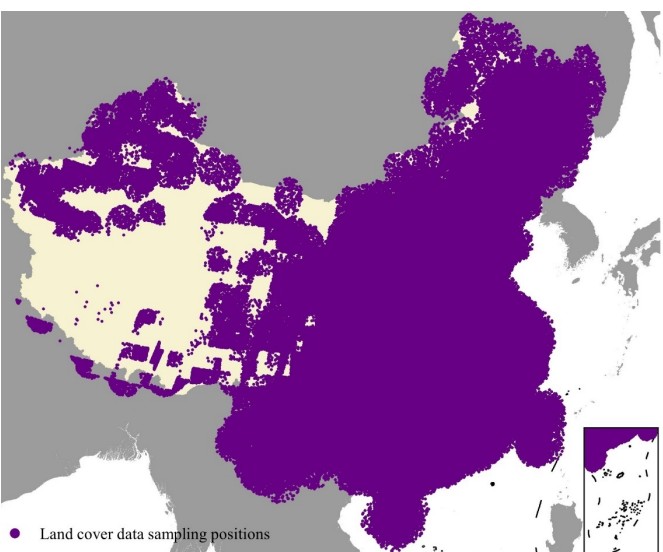

● Land cover data sampling positions

**Figure A1: The position of generated land cover samples. Base map © OpenStreetMap contributors 2023. Distributed under the**
**Open Data Commons Open Database License (ODbL) v1.0.**



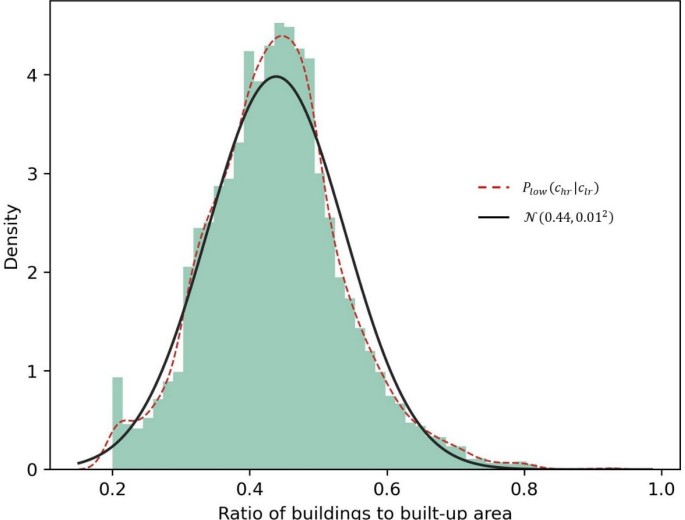

**Figure A2: The probability density distribution. The red dot curve is our statistical result based on the collected high-resolution survey data and the corresponding low-resolution built land cover data. The black curve represents the estimated Gaussian distribution with 0.44 as $\mu$ and 0.01 as $\sigma$. It is worth noting that our estimate is only derived from urban areas and is therefore biased.**
**However, as previously shown (Malkin et al., 2018), the impact of this bias is insignificant because of the robustness of the deep learning model in its modern design.**

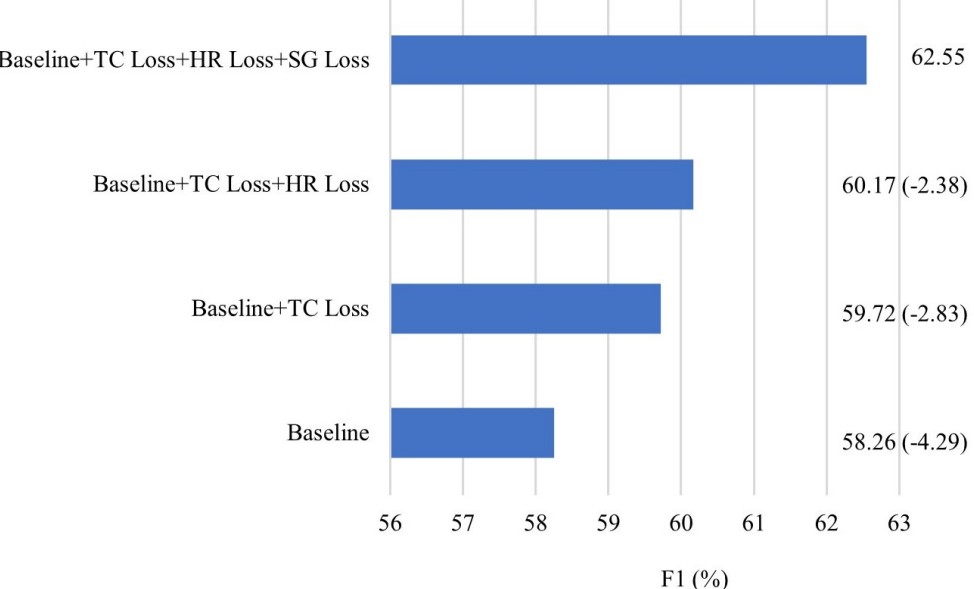

**Figure A3: Ablation study of our training method. The first row shows the F1-score performance of our best model including all the learning strategies, i.e., TC loss, HR loss and SG loss. The second row shows the best model without SG loss. The third row shows**
**the best model without HR Loss and SG loss. The last row is our baseline.**



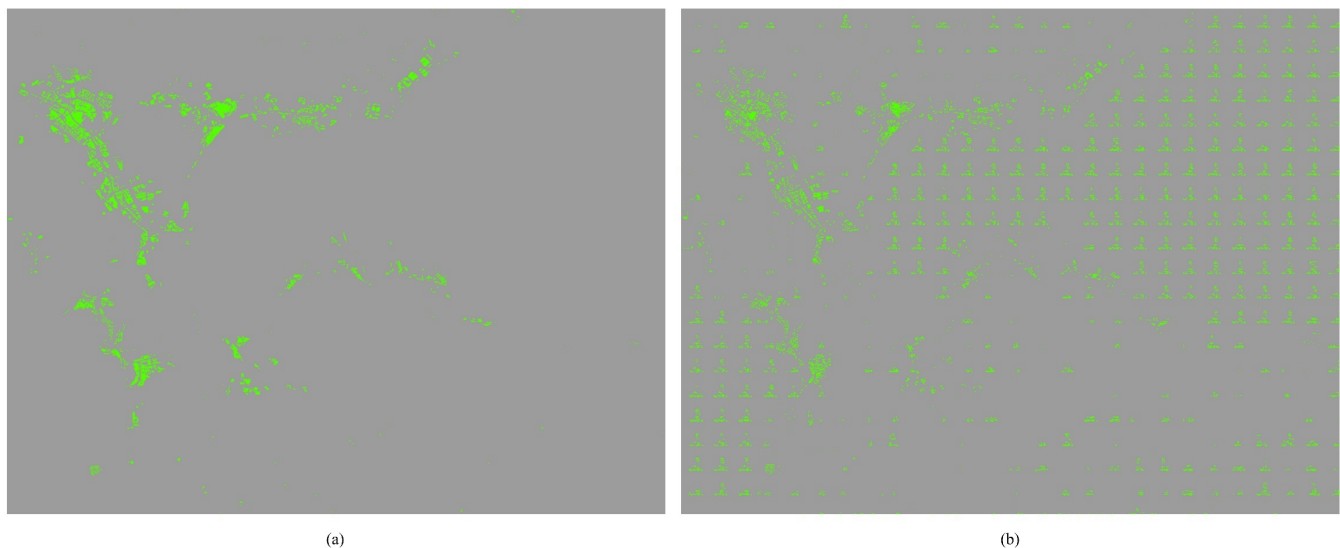

(a)                                                                 (b)

**Figure A4: An example of the importance of SG loss. (a) The result of our best model. (b) The result of our best model without SG loss supervision. Without utilizing SG loss, the model only converges on the limited collected survey rooftop data, which limits the generalization performance of the model over a large scale (e.g., national scale), resulting in unexpected false predictions.**

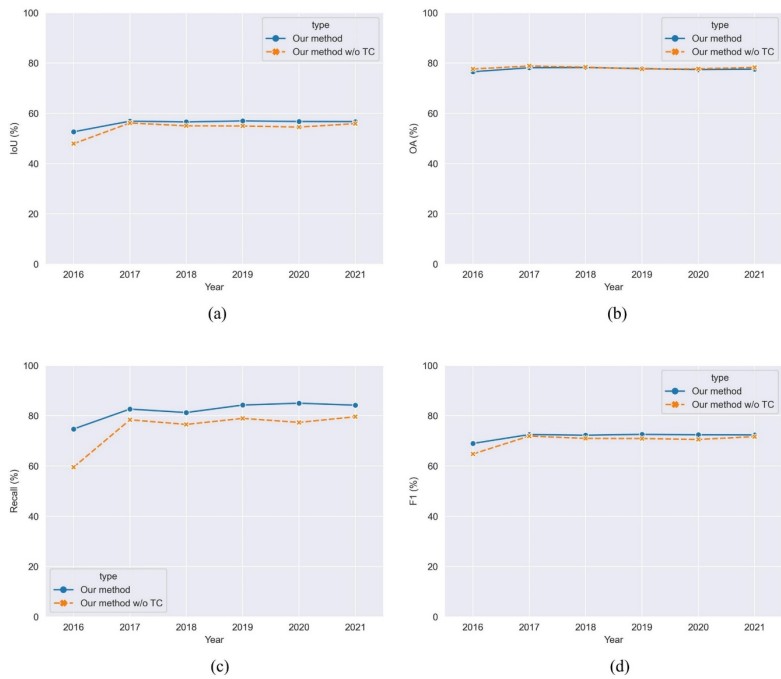


**Figure A5: Temporal accuracy in the old town of Beijing of our method and our method without TC loss. (a) IoU. (b) OA. (c) Recall. (d) F1.**



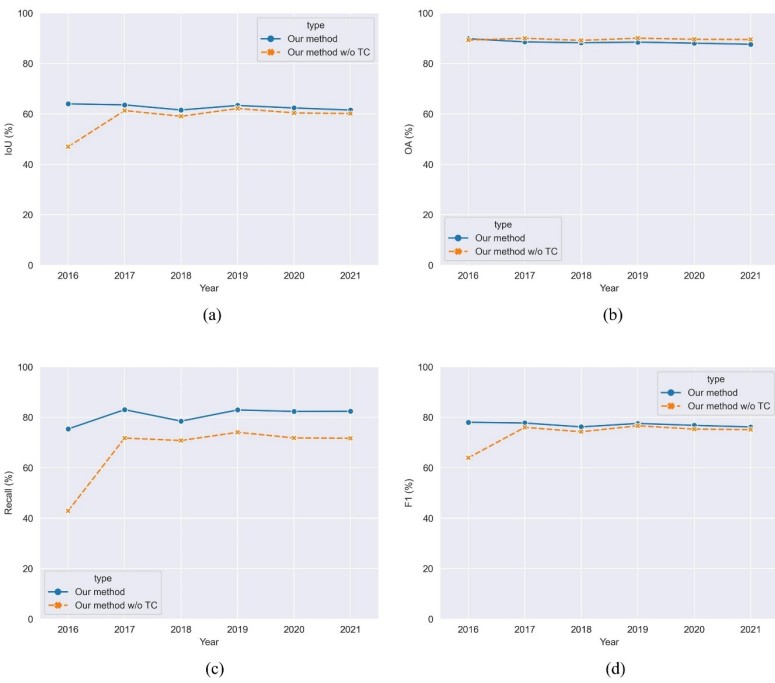

**Figure A6: Temporal accuracy in Hong Kong of our method and our method without TC loss. (a) IoU. (b) OA. (c) Recall. (d) F1.**


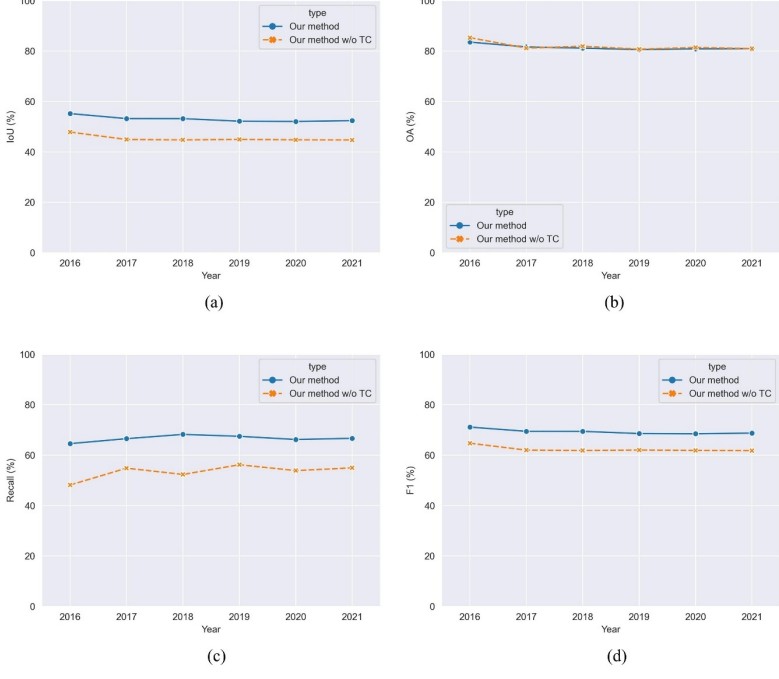

**Figure A7: Temporal accuracy in Macao of our method and our method without TC loss. (a) IoU. (b) OA. (c) Recall. (d) F**



## Appendix B: Tables

**Table B1: Statistics of building rooftop areas in the regions where the Sentinel-2 images are missing due to our cloud filter criteria. The image (cloud less than 20%) in 2021 is full coverage of China. Therefore, we utilize the CBRA results of 2021 as the benchmark to count the building rooftop area in the missing image region in each year (2016-2020) (the second column). In addition, we also calculate the proportion of missing building rooftop areas relative to the average area of building rooftops in China from 2016 to 2021 (89,826 $Km^2$) (the third column). It could be observed that the impact of the missing image is negligible.**

|  | Building rooftop area in the region where the Sentinel-2 image is missing | The proportion of the missing building rooftop area |
|---|---|---|
| 2016 | 704.89 $Km^2$ | 0.7847 % |
| 2017 | 70.93 $Km^2$ | 0.0790 % |
| 2018 | 36.82 $Km^2$ | 0.0410 % |
| 2019 | 7.55 $Km^2$ | 0.0068 % |
| 2020 | 5.90 $Km^2$ | 0.0066 % |

**Table B2: Descriptions of the built class in Dynamic World product, from its data report** (Brown et al., 2022).

|  | Description | Example |
|---|---|---|
| Built area | • Clusters of human-made structures or individual very large human-made structures.<br>• Contained industrial, commercial, and private buildings, and the associated parking lots.<br>• A mixture of residential buildings, streets, lawns, trees, isolated residential structures or buildings surrounded by vegetative land covers.<br>• Major road and rail networks outside of the predominant residential areas.<br>• Large homogeneous impervious surfaces, including parking structures, large office buildings, and residential housing developments containing clusters of cul-de-sacs. | • Cluster of houses, can include smalls lawns or small patches of trees can be included<br>• Dense villages, town, and cityscape (buildings and roads together)<br>• Clusters of paved roads and large highways<br>• Asphalt and other human-made surfaces |





**Table B3: The parameter list and our choice of the overall workflow.**

| Parameter | Description | Choice |
|---|---|---|
| Framework parameter | | |
| H, W, C | The height, width and channel of the input image patch. | 64, 64, 4 (i.e., RGB and NIR bands) |
| $r$ | The up-sampling factor. | 4 |
| $d$ | The dimension of the temporal representation. | 128 |
| $\tau$ | The temperature hyperparameter scaling the distribution of the TC loss. | 0.75 |
| N | The size of the memory bank maintaining the unrelated representation. | 16 |
| $\varepsilon = 10^{-6}$ | Constant term used to make the Focal Tversky Loss numerically stable. | 10-6 |
| $\gamma = 0.5$ | Focal term for shifting the model to converge to the easy samples (<1) or the hard samples (>1). | 0.5 |
| $\beta = 0.6$ | Penalty term for the model focusing on minimizing FP predictions (<0.5) or the FN predictions (>0.5). | 0.6 |
| $\alpha: \varphi: \omega$ | The ratio of the TC loss, HR loss and SG loss. | 1:200:5 |
| Training parameter | | |
| Max epoch | The number of passes a training dataset | 400 |
| Learning rate | The updating pace of the parameter | 0.00015 |
| Optimizer | The mechanism to adjust the parameter | Adam optimizer with default setting |
| Augmentations | Rules for creating various images to tackle overfitting. | Flips, rotate and colour jittering |
| Batch size | The number of samples processed before the parameters updated | 32 for high-resolution training set, 48 for low-resolution training set |








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
