# Peer review of "CBRA: The first multi-annual (2016-2021) and high-resolution (2.5 m) building rooftop area dataset in China derived with Super-resolution Segmentation from Sentinel-2 imagery"

_Earth System Science Data, 2023_

## Author Comment (AC1)

**Response to the Anonymous Referee #1**

General comment:

The manuscript presents the China Building Rooftop Area (CBRA) dataset, which provides national-scale pixel-level information on individual building rooftop distribution and multi-annual dynamics from 2016 to 2021. The authors proposed an interesting and novel method for extracting high-resolution production of building rooftop from Sentinel images that could potentially reduce data acquisition costs. The study is well-structured, and the results demonstrate characteristics and superiority over previous production. The paper could be a valuable contribution to the society of urban remote sensing in terms of both the novel methodology and production. However, some revisions are necessary before accepting it for publication.

Response:

We express our gratitude to the reviewer for acknowledging our research and offering valuable suggestions. In the following section, we present a detailed response to each comment provided. The reviewer's comments are presented in black font, while our responses are presented in blue font. The revised manuscript is presented in red color, and our modifications are highlighted in yellow.

Major comments 1:

The one of the key contributions in this manuscript is the usage of a spatial generalization (SG) loss to increase the generalization capacity of a larger geographical region. However, as shown in figure 9, 12, 13, there exist some block-like result. It seems the SG loss could not perform well in these regions. Are they all caused by a resolution of less than 2.5 meters so the model can't distinguish them? The authors should provide further justification of the importance of the SG loss, including its superiority compared to directly utilizing cross-entropy.

Response:

Thank you for your valuable comments.

The block-like results are mainly caused by the buffer-like effect of a few pixels at the building edges caused by our approach, which is common in the building extraction practice (Ding et al., 2021; Zorzi et al., 2021; Guo et al., 2022; Liu et al., 2022). We have included examples of this effect in Fig. C1, where we present an example of boundary localization results using the Unet++ method (Zhou et al., 2019). Even in very high-resolution images (0.3 m), the building boundary extraction results exhibit an uncertain offset of several pixels.

[Figure]

|     (a)     |     (b)     |     (c)     |     (d)     |

**Figure C1: An example of the ambiguity of building edges from Inria building dataset (0.3 m) by the Unet++ method (Zhou et al., 2019). (a) The input image (0.3 m). (b) The ground truth. (c) The extraction result. (d) The building boundary, where red is ground truth and blue is prediction, respectively. It can be from (c) that the building boundary extraction results still shows with an offset of several pixels even from the very high-resolution images (0.3 m).**

In densely residential areas, the aforementioned ambiguity is further compounded, as demonstrated in Fig. C2. When buildings are close (e.g., less than 6-7 m, i.e., 2-3 pixels at 2.5 m resolution image), our method may not be able to effectively distinguish between them. It is a common phenomenon in building extraction task, but the previous work mainly uses very high-resolution images (less than 1 m), making their results seem better than ours.

[Figure]

|     (a)     |     (b)     |

**Figure C2: Two examples of blob-like areas and the measured distances between adjacent buildings. (a) Densely residential area (101.302089ºE, 21.298532ºN). (b) Relatively discrete residential area (121.634662 ºE, 31.746674 ºN). Imagery © 2023 Maxar Techonlogies.**

As for the SG loss, as discussed in Sect. 6.1 "Importance of the spatio-temporal aware learning", using SG loss as an auxiliary loss could greatly improve the performance than only using cross entropy loss (2.38% improvement in F1 score). The SG loss mainly helps us to obtain more weakly supervised information in rural areas, thus improving the generalization performance of the model. However, due to the spatial resolution limitation, some regions are more seriously adhered.

Therefore, we now strengthen the statement of limitation (Sect. 6.2) in the revised manuscript (Fig. 19 in the revised manuscript is the Fig. C2 above), from line 608 to 614 on page 30:

Although our STSR-Seg framework is scalable, allowing larger areas to be monitored (e.g., national scale). there remain some limitations to our approach. Specifically, the segmentation results for densely populated residential areas may present certain rooftops as a single block, rather than individual buildings. Our analysis suggests that this occurrence is primarily due to the resolution of the results, which is 2.5 m. Furthermore, the semantic segmentation technique utilized in the approach may introduce some uncertainty at the edges of buildings, resulting in additional pixels, up to three pixels, at the boundary. Consequently, up to 7.5 meters of buffering may occur, exacerbating the problem of building adhesion. Examples of this issue are presented in Fig. 19

References:

Ding, L., Tang, H., Liu, Y., Shi, Y., Zhu, X. X., & Bruzzone, L. (2021). Adversarial shape learning for building extraction in VHR remote sensing images. IEEE Transactions on Image Processing, 31, 678–690.

Zorzi, S., Bittner, K., & Fraundorfer, F. (2021). Machine-learned regularization and polygonization of building segmentation masks. 2020 25th International Conference on Pattern Recognition (ICPR), 3098–3105.

Guo, H., Du, B., Zhang, L., & Su, X. (2022). A coarse-to-fine boundary refinement network for building footprint extraction from remote sensing imagery. ISPRS Journal of Photogrammetry and Remote Sensing, 183, 240–252.

Liu, Z., Tang, H., & Huang, W. (2022). Building Outline Delineation From VHR Remote Sensing Images Using the Convolutional Recurrent Neural Network Embedded With Line Segment Information. IEEE Transactions on Geoscience and Remote Sensing, 60, 1–13.

Zhou, Z., Siddiquee, M. M. R., Tajbakhsh, N., & Liang, J. (2019). Unet++: Redesigning skip connections to exploit multiscale features in image segmentation. IEEE Transactions on Medical Imaging, 39(6), 1856–1867.

Major comments 2:

While the SG loss might be reasonable for regions with missing high-resolution supervised information, the authors assign a "built" land cover type to each building rooftop reference as described in line 260, which might potentially confuse the original high-resolution information in my personal viewpoint. Therefore, a detailed description of the pipeline is required when both high-resolution and low-resolution references are available.

Response:

Thank you for your valuable comments.

As shown in Fig.6 of the manuscript, the SG loss is used to provide additional supervised information with coarse resolution (i.e., the "built-up" area in Dynamic World product) to ensure that the model can effectively generalize to a larger geographic region, especially in the absence of

reliable high-resolution building references (i.e., the building rooftop from Tian-di Map). We also find that using low-resolution labels in the presence of high-resolution labels as well improves the accuracy of the model, as shown in the following table:

**Table C1: The impact of collaborative training with high-resolution references and low-resolution references. When using the SG loss, there are two strategies. One involves the use of low-resolution references solely in the absence of high-resolution supervised information, while the other entails training both simultaneously (i.e., utilizing low-resolution information for supervision at the same location even in the presence of high-resolution references). Our empirical results indicate that the second approach, i.e., collaborative training, can achieve superior accuracy compared to the former.**

|  | IoU | OA | Recall | F1 |
|---|---|---|---|---|
| STSR-Seg w/o collaborative training | 45.15 | 82.73 | 74.51 | 62.21 |
| STSR-Seg w/ collaborative training | 45.51 | 82.85 | 74.66 | 62.55 |

We add more details about it in Section 6.1 (Table B4 refers to the Table C1 above), from line 583 to line 584 on page 29 :

……of training resources, and therefore greatly improves the accuracy of our data-driven method (+2.38% in terms of F1 score). Even when high-resolution references are available, incorporating low-resolution land-cover information in the training process through collaboration as supervised information is found to be beneficial (Table B4). In addition, ……

Major comments 3:

By comparing with the 90 cities BRA data, the authors claimed that the results of the 90 cities BRA data are spatially inconsistent than the CBRA. It is necessary to ensure that the supervised data is similar between the CBRA and the 90 cities BRA to rule out any differences that could be causing this inconsistency.

Response:

Thank you for your valuable comments.

According to the technical report of 90 cities BRA data, the author did not reveal more details about the training samples. This makes it difficult to accurately determine any differences between their training samples and ours. However, we believe the Google Earth Satellite imagery potentially have geographical offsets, particularly in regions where high-rise buildings are present and display shifts, as illustrated in Fig. C3. This error can be attributed to variations in the topography's elevation and is less apparent in the low-rise buildings. In all, it is believed that the spatial inconsistency is primarily attributed to image shifts.

[Figure]

(a)             (b)

**Figure C3: Two examples about the stitching part of GES imagery. (a) The high-rise buildings. (b) The low-rise buildings. It could be seen that the offset is more obvious in the high-rise buildings than that in the low-rises. Imagery © 2023 Maxar Techonlogies.**

Major comments 4:

The paper highlights the limited availability of datasets covering the entire China. However, there exist some BRA datasets derived from sub-metric aerial images that cover specific regions. It would be valuable to show a comparison of the CBRA dataset with other small yet region-specific BRA datasets to gain more insights.

Response:

Thank you for your valuable comments.

We have attempted to procure regional building datasets to conduct an in-depth analysis of our product. However, it is regrettable that most of the locally-distributed building datasets in China are labeled manually and lack geographic information, such as coordinate systems, necessary for unbiased comparison (Li et al., 2022). Therefore, it is challenging to locate a regional dataset to compare our product fairly.

Nonetheless, our manuscript showcases the results of comparing our product with the 90-cities-BRA, comprising of 245, 458 independent samples. Additionally, we have conducted trend analysis using CLCD, GAIA, and Dynamic World product, illustrating the temporal consistency of the CBRA.

References:

Li, J., Huang, X., Tu, L., Zhang, T., and Wang, L.: A review of building detection from very high resolution optical remote sensing images, GIsci Remote Sens, 59, 1199–1225, 2022.

Major comments 5:

The multi-annual dynamic world product is presented as a probability map. Therefore, it is necessary to provide a more detailed technical description of the threshold, which might be used to binarize it in a meaning way.

Response:

Thank you for your valuable comments.

In Section 4.4, we use 0.2 to binarize the Dynamic World product and use it to intersect with our result to filter the potential false predictions. To find a reliable threshold, we need to make sure the binarized map would not filter out the correct predictions. Therefore, two considerations are involved:

(1) First, with a threshold interval of 0.05, we binarize the dynamic world product and calculate its recall on the building rooftop area data we collected from 52 cities, and the results are shown in Fig. C4. It can be found that recall reaches 0.99 when the threshold value is 0.3, which shows that using a threshold value of 0.3 and below ensures that the correct results are not filtered out incorrectly in urban areas.

(2) Secondly, since the above reasonable thresholds are calculated based on urban samples, we also need to consider rural areas. Due to the lack of reliable building distribution data for rural areas, we judge the thresholds by visual observation of the images, and finally we consider 0.2 as a more robust threshold because it does not filter out our correct prediction results.

[Figure]

**Figure C4: The recall curve for the threshold selection. The threshold interval of 0.05 was used to binarize the Dynamic World product, and it was compared with the building distribution data we collected from 52 cities in China to calculate recall respectively. When the threshold is 0.3, recall reaches 0.99, indicating a threshold value of 0.3 and below ensures that the correct results are not filtered out incorrectly in urban areas. As for the rural area, due to the lack of reliable building rooftop annotations, we judge the thresholds by visual observation of the images. Finally, we consider 0.2 is the best option.**

We integrate the relating explanations into the revised manuscript (Fig. A3 is the Fig. C4 above), from line 383 to 386 on page 16.

…… The built area provided by Dynamic World is a possibility estimation ranging from 0-1. A low threshold of 0.2 is utilized to distinguish between built-up and unbuilt areas, as this threshold

==does not filter out correct prediction results (further discussions in Fig. A4).==

Minor Comments 1:

The introduction and background sections have some repetitive descriptions. For example, the second paragraph could be shortened for there already has a detailed description in Section 2.2.

Response:

Thank you for your comment. The second paragraph in the introduction is shorten in the revised manuscript.

Minor Comments 2: line 60: please provide the source of the statistics about the urbanization rate and the population structure of China.

Response:

Thank you for your comment. The source of the statistics is from the National Bureau of Statistics of China, we now add this reference into the manuscript.

Minor Comments 3:

line 159: "apple" should be "apply"

Response:

Thank you, confirmed and corrected in the revised manuscript.

Minor Comments 4: l

line 220: please unify the description of "arcgis"

Response:

Thank you, confirmed and corrected in the revised manuscript.

Minor Comments 5:

Figure 5: what is the meaning of the four rectangle and arrow in subfigure (C)?

Response:

Thank you, it is a schematic diagram about the sliding window for obtaining the Sentinel-2 images

Minor Comments 6:

Figure 6: the direction of blue arrow and red arrow in the legend should be the same.

Response:

Thank you, the Fig. 6 is updated with a new legend in the revised manuscript.

Minor Comments 7:

line 268: correct "red backward arrow" to "red arrow"

Response:

Thank you, confirmed and corrected in the revised manuscript.

---

## Author Comment (AC2)

**Response to the Anonymous Referee #2**

General comment:

The authors used the STSR-Seg method to develop a novel BRA dataset covering 2016-2021 with a temporal interval one-year, it has been demonstrated to have better performance than Zhang et al. (2022) results and covers the whole China (rural and urban areas), with an overall accuracy of 82.85%. The manuscript has been well-written and clearly organized. However, I still have several concerns about the current manuscript as follows

Response:

We express our gratitude to the reviewer for acknowledging our research and offering valuable suggestions. In the following section, we present a detailed response to each comment provided. The reviewer's comments are presented in black font, while our responses are presented in blue font. The revised manuscript is presented in red color, and our modifications are highlighted in yellow.

Comment 1: The generating training samples should be greatly improved. For example, are the sample size of 200 and standard deviation of 150 reasonable? The land-cover label of each training coordinate came from which dataset (the spatial resolution of the coordinate represent 10 m spatial resolution?). As for the reference year, why you randomly assigned during 2016-2021? And you stated '*we further rebalance the gathering samples to make sure that the built type is the majority by thresholding*', how to achieve the goal?

Response: Thank you for the reviewer's comment.

Before responding your comments. We would like to restate our motivations of this generating process to better make a response.

Our method (STSR-Seg) needs two types of reference data (i.e., supervised information) in the training process. One is the accurate building rooftop data (i.e., vector data collected from Tiandi-Map). However, it only covers 52 cities and the supervised information is not sufficient to support our mapping needs for building rooftops for multiple years across the whole country. Therefore, we collected another data for weak supervision information, i.e., coarse resolution (10m), but full coverage, as well as multi-year built-up area data. This data was collected from the Dynamic World product (Dynamicworld, 2022).

When generating training samples, we need to consider two issues.

(1) Firstly, we need to make it as inclusiveness of built-area as possible as we could. Therefore, we sampled in the vicinity of the administrative point in China, and used a Gaussian distribution for sampling. In our experiments, we found the standard deviation of 150 and sampling size of 200 are the best for covering the possible regions (Fig. A1 in the manuscript of page 33).

(2) Second, we need to consider the engineering efficiency and resources for local storage. Considering that the amount of training data used in the current state-of-the-art remote sensing large

models in deep learning field is one million (Wang et al., 2022), we want to make our training sample size approach this order of magnitude. There are 2, 844 basic points used to determine the sampling range. So, the amount of the training data size (N) is:

$$N = 2844 \times 200 \times t,$$

where $t$ is the number of the sampling year. In this paper, to approach the magnitude of one million, we used 3 for $t$, i.e., "randomly assign three reference years over 2016-2021." Although more sampling years may improve the accuracy (due to more temporal samples), the resulting computational effort is also larger. Based on the current amount of data, it takes us about 1 month to finish the model training only once using our local server. We sometimes reset the hyper-parameters to re-train the model, and the actual time is longer.

Finally, not all samples should be used, because some of them have few pixels in the built-up area and result in a long-tail distribution problem (as shown in Fig. A2 in the manuscript, some samples are even in the sea area). So, we need to perform a second filtering, i.e., filtering by the built-up area pixel proportion. Specifically, those with built-up area pixel percentage less than 10% are discarded when computing the loss.

In all, (1) the sample size and the standard deviation are set to best cover the entire built-up area of China (Fig. A2). (2) The land-cover label of each training coordinate came from the Dynamic World product (built-up area). (3) The number of randomly sampled years is used to control the total number of training samples, and 3 is the optimal option based on the previous work and the upper limit of our computational resources. (4) Further filtering is done by a 10% threshold to eliminate samples that do not contain built-up areas and potentially erroneous samples.

We integrate the relating explanations into the revised manuscript, from line 239 to 263 on page 11 and 12:

…… As described in Sect. 3, the reference data for training consists of both the high-resolution building rooftop in 47 cities from Tiandi-Map and the low-resolution land cover data from the Dynamic World product. For the rooftop data, we can easily pair them with the Sentinel-2 imagery obtained in the same location and time…….

The number of sampling coordinates and standard deviation employed in the heuristic sampling strategy is based on off-line experiments, which thoroughly cover the potential urban areas of China (Fig. A2). It is important to note that only three years are randomly sampled from 2016-2021 to avoid increasing the dataset and imposing an unmanageable computational burden. Through this approach, the land cover training samples, covering both urban and rural scenes and ranging for various years, are easily and automatically gathered.

Finally, the generated samples may still exhibit redundant information, necessitating their further filtration. Specifically, those samples containing fewer built-up area pixels (i.e., below 10%) are discarded. In all, we obtain two sets for model training……

References:

Dynamicworld: https://dynamicworld.app/, last access: 24 November 2022.

Wang, D., Zhang, Q., Xu, Y., Zhang, J., Du, B., Tao, D., and Zhang, L.: Advancing plain vision transformer towards remote sensing foundation model, IEEE Transactions on Geoscience and Remote Sensing, 2022.

Comment 2: The authors used the spatiotemporal learning method to achieve the aim of downscaling and independently generate the CBRA map in each year, but the block effect is still obvious in your results. To further demonstrate the feasibility of the proposed method, can you use the Landsat image (30 m) as an example to generate the CBRA at 2.5? I believe it would make more sense (only a suggestion).

Response: Thank you for the reviewer's suggestion.

We feel that using Landsat data does not necessarily lead to better results. This is due to our methodology's reliance on a framework that integrates super-resolution and semantic segmentation techniques. We believe that the accuracy of the model's output results is contingent on the alignment between the original image resolution and the resolution of the resulting data (Shermeyer & Van Etten, 2019). As such, the greater the similarity between the two resolutions, the higher the model's accuracy. In our work, we established an output resolution of 2.5 meters. The Sentinel-2 data, with its coverage of the entire Chinese territory over many years and a resolution of up to 10 meters, represents the optimal data source for our study. In contrast, the Landsat image, with a resolution of 30 meters, may not enhance the model's accuracy due to the significant discrepancy between its resolution and the specified output resolution of 2.5 meters.

References:

Shermeyer, J., & Van Etten, A. (2019). The effects of super-resolution on object detection performance in satellite imagery. Proceedings of the IEEE/CVF Conference on Computer Vision and Pattern Recognition Workshops, 0.

Comment 3: In section 4.4, the authors used a lot of thresholds (0.5 and 0.2) according to their prior knowledges, however, as for a national-scale mapping method, the authors should give the analysis of why these thresholds are reasonable.

Response: Thank you for the reviewer's comment.

There are two thresholds involved. One is the threshold used to binarize the probability map (value range between 0 and 1) of the output from our STSR-Seg model. The threshold is 0.5 here since it is used to discriminate the two classes, i.e., building or not, during model training. This is a common setting based on the previous research on semantic segmentation (Liu and Tang, 2023). We followed this setup directly.

Another threshold is about thresholding the Dynamic World product. The Dynamic World product provides the probability estimation (value range between 0 and 1) of the built-up area. In Section 4.4, we want to use this product to make an intersection with our result to remove the potential false predictions. Therefore, we need to binarize the Dynamic World by a threshold. This threshold was determined though the following way:

(1) First, with a threshold interval of 0.05, we binarize the dynamic world product and calculate its recall on the building rooftop area data we collected from 52 cities, and the results are shown in Fig. C1. It can be found that recall reaches 0.99 when the threshold value is 0.3, which shows that using a threshold value of 0.3 and below ensures that the correct results are not filtered out incorrectly in urban areas.

(2) Secondly, since the above reasonable thresholds are calculated based on urban samples, we also need to consider rural areas. Due to the lack of reliable building distribution data for rural areas, we judge the thresholds by visual observation of the images, and finally we consider 0.2 as a more robust threshold because it does not filter out our correct prediction results.

[Figure]

Figure C1: The recall curve for the threshold selection. The threshold interval of 0.05 was used to binarize the Dynamic World product, and it was compared with the building distribution data we collected from 52 cities in China to calculate recall respectively. When the threshold is 0.3, recall reaches 0.99, indicating a threshold value of 0.3 and below ensures that the correct results are not filtered out incorrectly in urban areas. As for the rural area, due to the lack of reliable building rooftop annotations, we judge the thresholds by visual observation of the images, and finally we consider 0.2 is the best option.

We now integrate the relating explanations into the revised manuscript (Fig. A4 is the Fig. C1 above in the revised manuscript), from line 379 to 386 on page 16:

……removing the zero padding. (5) A threshold value of 0.5 is used to differentiate between candidate foreground pixels (i.e., building rooftop) and background pixels, following common practice (Liu and Tang, 2023).

…… The built area provided by Dynamic World is a possibility estimation ranging from 0-1. A low threshold of 0.2 is utilized to distinguish between built-up and unbuilt areas, as this threshold does not filter out correct prediction results (further discussions in Fig. A4).

References:

Liu, Z. and Tang, H.: Learning Sparse Geometric Features for Building Segmentation from Low-Resolution Remote-Sensing Images, Remote Sens (Basel), 15, 1741, 2023.

Comment 4: In term of temporal-optimization, authors used the temporal checking method (proposed by Li et al. (2014)) for optimize the multitemporal CBAR results. Did the authors consider whether classification errors would be introduced into the optimized results?

Response: Thank you for the reviewer's comment.

We have considered the building's characteristics when utilizing the temporal checking methodology. We assumed that the building would not undergo continuous changes in its state within a 3-year period, such as construction-demolition-construction or demolition-construction-demolition. Therefore, we limited the sliding window to 3 and did not perform iterative filtering by expanding window as in the approach proposed by Li et al. (2015). The filtering method we used was the most conservative, and any errors it introduces should be minimal compared to the accuracy improvement it provides.

We now add our consideration into the revised manuscript, from line 389 to 395 on page 17:

Due to the possible random bias of our method in locating the boundary of building rooftops (outlined in Sect. 6.2), inconsistencies in identification results over time for the same building may occur. To address this issue, a temporal homogeneity check approach has been implemented. Specifically, it is assumed that a building's state does not undergo continuous change over three successive years. Building upon the method proposed by Li et al, (2015), a $3 \times 3$ sliding window is utilized to determine the final pixel value through majority voting, as illustrated in Fig. 8. This procedure ensures that the results for adjacent years are comparable. However, for the edge year like 2016 and 2021, they are not checked due to the lack of temporal information.

References:

Li, X., Gong, P., and Liang, L.: A 30-year (1984–2013) record of annual urban dynamics of Beijing City derived from Landsat data, Remote Sens Environ, 166, 78–90, 2015.

Comment 5: As for the validation, I think the comparisons with impervious surfaces products (CLCD, GAIA) might not make sense. Instead, they should pay more attention on the comparison with CBA dataset and add more BAR dataset (such as regional dataset) as comparison dataset.

Response: Thank you for the reviewer's comment.

In our study, we have employed a rigorous approach to validate the proposed dataset CBRA

from two perspectives. Firstly, we have compared CBRA with a previously published dataset, namely 90-cities-BRA (Zhang et al., 2022), which is the only comparable and available dataset up to now. Both datasets were generated using remote sensing imagery through automated procedures, and we have reported the comparison results in Table 4 of the manuscript.

Secondly, since multi-year building rooftop data for China were not previously available, we have employed other relatively coarse resolution impervious surface or built-up area datasets, namely CLCD, GAIA, and Dynamic World, to establish trends and validate our results. Specifically, we have calculated the correlation coefficients between different datasets for the percentage of foreground pixels in a 0.10° x 0.10° grid, as presented in Fig. 15 of the manuscript. This validation approach has been utilized in other studies as well (Yang and Huang, 2021) and is considered a reasonable method to establish the validity of our dataset.

Unfortunately, we could not find any multi-annual regional datasets that could be used for fair comparison with our proposed dataset. While there are some building datasets available in the public domain, most of these datasets are created through human-labelling for competitions or method development, and do not contain crucial geo-information such as the coordinate system (Li et al., 2022). Therefore, in China, the only comparable dataset available is the 90-cities-BRA dataset.

References:
Zhang, Z., Qian, Z., Zhong, T., Chen, M., Zhang, K., Yang, Y., Zhu, R., Zhang, F., Zhang, H., and Zhou, F.: Vectorized rooftop area data for 90 cities in China, Sci Data, 9, 1–12, 2022.
Yang, J. and Huang, X.: The 30 m annual land cover dataset and its dynamics in China from 1990 to 2019, Earth Syst Sci Data, 13, 3907–3925, 2021.
Li, J., Huang, X., Tu, L., Zhang, T., and Wang, L.: A review of building detection from very high resolution optical remote sensing images, GIsci Remote Sens, 59, 1199–1225, 2022.

Comment 6: The quantitative analysis in Section 5.1.1 is interesting, can you give the reasons why the BAR dataset achieved higher accuracy in rural areas than that of the urbans.
Response: Thank you for the reviewer's comment.

We believe this comes mainly from the difference between the urban test sample and the rural test sample we use. For urban areas we used a reliable survey-based sample for testing, containing 245,458 buildings. These samples are very accurate and detailed.

However, since it is still very challenging to find reliable test samples in rural areas, we collected 33,736 building rooftops from Open Street Map and manually corrected them using high-resolution images. We labeled more obvious buildings due to limitations in image resolution. Consequently, the rural test sample was relatively less difficult for the model to identify compared to the urban sample. The report accuracy shows higher accuracy in rural areas than in urban areas. The same conclusion was drawn for GHSL in our benchmark. Its recall value in rural areas was also

higher than in urban areas.

In the original manuscript, we mainly conducted product-to-product comparisons without delving into the differences between our products in urban and rural areas. We concur it is necessary to illustrate this. Therefore, we added a description on this aspect to Section 5.1.1 in the revised manuscript, from line 448 to 451 on page 19:

……However, the CBRA is very close to it (78.94%), with a gap of only 1.95%, indicating its reliability in predicting building rooftops in rural areas. Considering the varieties of urban and rural test samples, it should be mentioned that the presented results in Table 4 intend to compare product-to-product, rather than to demonstrate performance differences between urban and rural areas.

Comment 7: The figure 10 illustrated that the CBAR dataset has obvious advantages than BRA dataset in their previous study. However, why the BRA has high geometry accuracy in top left corner of the Figure 10c and suffers obvious offset in the central areas (red circle) over such local area.

Response: Thank you for your comment.

The compared data (i.e., the 90-cities-BRA in Fig. 10c) is produced from Google Earth Satellite (GES) imagery. When acquiring larger scale GES images, stitching of images from multiple sensors is generally required. When stitching, due to the elevation change of the topology, there will be large errors at the stitching regions, especially in places with large height fluctuations (e.g., high-rise buildings). Examples are shown in the below, and the images are captured from Google Earth Pro software.

[Figure]

(a)                                                                          (b)

**Figure C2: Two examples about the stitching part of GES imagery.(a) The high-rise buildings. (b) The low-rise buildings. It could be seen that the offset is more obvious in the high-rise buildings than that in the low-rises. Imagery © 2023 Maxar Techonlogies.**

In Fig 10 of our manuscript, the red circle designates the high-rise buildings, highlighting that the bias is more prominent than in the surrounding low-rise buildings. Given that the 90-cities-BRA manuscript by Zhang et al. (2022) did not provide additional information on the assessment of image quality, we speculate that their errors may have arisen due to the aforementioned reason.

Details regarding this phenomenon are introduced in the revised manuscript (Fig. A1 is the Fig. C2 above), line 159 on page 6:

However, the GES images are collected from various kinds of high-resolution satellites, and have two potential problems when applied to large-scale mapping: (1) inconsistent geographical offset (illustrated in Fig. A1), and (2) inconsistent acquisition time (e.g., the image is obtained from various satellite sensors with different acquisition times) which results in spatio-temporal inconsistency in the generated product.

References:

Zhang, Z., Qian, Z., Zhong, T., Chen, M., Zhang, K., Yang, Y., Zhu, R., Zhang, F., Zhang, H., & Zhou, F. (2022). Vectorized rooftop area data for 90 cities in China. Scientific Data, 9(1), 1–12.

Comment 8: The temporal analysis in Section 5.2 should be strengthen, as the increased/decreased BRA is great small than the stable BRAs, combining the stable and changed BRAs for analyzing the accuracy metrics cannot illustrate the performance of proposed method in the temporal dimension. For example, you can use the changed validation points to analyze the changed CBRAs.

Response: Thank you for the reviewer's comment.

We appreciate the significance of using updated validation points in our research; however, implementing such a validation process has posed several challenges.

Firstly, identifying validation samples from publicly available Sentinel or Landsat imagery through manual labeling is feasible, with the exception of building rooftops, where the task is complicated by the unavailability of year-by-year high-resolution images from 2016-2021.

Secondly, although we can obtain some imagery from Google Earth, the inability to interact with commonly used GIS software, such as ArcGIS or QGIS, makes human labeling difficult. Additionally, Google Earth lacks images from certain years, further restricting our ability to carry out the labeling process.

Thirdly, we have been unable to locate any annual building rooftop or footprint datasets, including regional data, thereby hindering our ability to perform thorough testing of the CBRA using appropriate samples.

Therefore, in the original manuscript (Section 5.2), we assume that the buildings in the old town area of Beijing, Hong Kong and Macao are stable, and we use these samples to test our annual performance. Besides, we calculate the correlation coefficient of the CBRA and other annual products to check the trend consistency.

It is important to note that the assumption of building stability may not be entirely accurate, and we acknowledge that this may introduce some level of uncertainty in our validation process. However, given the limitations outlined earlier, we believe that utilizing the stable buildings in these areas was the most reasonable approach to validate our model's performance annually.

Comment 9 #1: Figure 18 is interesting, and two enlargements intuitively show the good performance of the CBAR. I suggest the authors added the high-resolution imagery over two enlargements to make the analysis more intuitive.

Response: Thank you for your comment.

Two enlargements are shown below (Fig. C3 and C4), which contain the demolition process and construction process between 2016 and 2021. Fig. C4 is a zoomed part of Fig. 18(b) in the manuscript. It shows the removal of the block consisting shack houses. Figure C5 is the zoomed part of Fig. 18(c) in the manuscript, it shows the construction of the modern apartments.

To increase the visibility, we pick the high-resolution images with high quality and large scale in the Google Earth Pro software. Owing to their magnified dimensions, they could not be accommodated in the primary body of the manuscript and were therefore included in the Appendix, and add some descriptions about them in the manuscript (Fig. A5 and A6 in the revised manuscript refer to the Fig. C3 and C4 in the author response), line 550 on page 25 :

……in the rural area, and the establishment of buildings (e.g., apartments) in the urban area, respectively. More comprehensible references about the building change can be found in Fig. A5, A6 and A7.. ……

[Figure]

(a)                                        (b)

**Figure C3: A zoomed result of the demolition process with high-resolution images (116.387674ºE, 39.766427ºN). (a) The identified building rooftop area of 2016. (b) The identified building rooftop area in 2021. Imagery © 2023 Maxar Techonlogies.**

[Figure]

(a)                 (b)

**Figure C4: A zoomed result of the construction process with high-resolution images (113.037558ºE, 23.014343ºN). (a) The identified building rooftop area of 2016. (b) The identified building rooftop area in 2021. Imagery © 2023 Maxar Techonlogies.**

Comment 9#2: In addition, why the North China Plain (especially in Henan province) shows such a marked increase? An enlargement in Henan should be added.

Response: Thank you for your comment.

The North China Plain mainly contains three provinces that contain Hebei, Henan, Shandong provinces and two municipality Beijing and Tianjin. In the previous research conducted by Gong et al, (2019), the top urban expansion provinces of China are Shandong, Jiangsu, Hebei, Guangdong, and Henan by 2017. The results of our analysis show the consistent trend, i.e., the roof area of buildings in the North China Plain, which is composed of Shandong, Hebei and Henan, produced a large increase from 2016-2021.

Regarding Henan province, we collected statistical data from the National Bureau of Statistics of China on the floor space of commercial and residential buildings sold between 2016 and 2021. Our analysis showed that Henan ranked fifth in the country in terms of commercial and residential building floor space sold, while Shandong ranked third. Although buildings in China are primarily tall buildings such as apartments, this indicator is related to building rooftop area and thus provides insight into the growth of building construction in Henan.

**Table C1: The top 5 change of sold floor space of commercialized and residential buildings (10000 sq.m) between 2016 and 2021 (from National Bureau of Statistics of China)**

| Province | Change of floor space of commercialized buildings sold between 2016 and 2021 | Change of Floor Space of Residential Buildings Sold between 2016 and 2021 | Total change |
|---|---|---|---|
| Sichuan | 4392.44 | 3028.05 | 7420.49 |
| Jiangxi | 2984.37 | 2540.71 | 5525.08 |
| Shandong | 2482.97 | 2033.47 | 4516.44 |
| Jiangsu | 2589.73 | 1703.84 | 4293.57 |
| Henan | 1970.92 | 2121.70 | 4092.62 |

Furthermore, the CBRA dataset we used includes various types of buildings, including residential, commercial, industrial, and agricultural. Henan is a major province for agriculture and industry, with the third population of 988 million in 2021 in China, making it more likely that building activities will be more frequent and have a greater impact on building rooftop area.

Lastly, we examined high-resolution images provided by Google satellite imagery alongside the CBRA results but did not observe any distinctive features of building change, such as a significant increase in a particular type of building. However, we noted that Henan Province is located in a plain with numerous human settlements, and each settlement experiences changes in buildings that contribute to the overall increase in the rooftop area of buildings in the province.

It is important to note that we did not conduct a more detailed analysis of the building change process as it would require additional methodologies or auxiliary data beyond the scope of this manuscript, Therefore, our focus was mainly on the overall changes in building rooftop area across China and major city clusters. The following Fig. C5 is an enlargement of the Henan province, and two examples of the new construction buildings between 2016 and 2021.

We add this figure to the revised manuscript (Fig. A7 is the Fig. C5 in the author response), line 550 on page 25:

……in the rural area, and the establishment of buildings (e.g., apartments) in the urban area, respectively. More comprehensible references about the building change can be found in Fig. A5, A6 and A7.. ……

[Figure]

**Figure C5: Enlargement of the Henan province of China. (a) The spatial distribution of the annual change of building rooftop area over the period of 2016-2021. The area fraction is aggregated within the $0.10° \times 0.10°$ spatial grid (base map © OpenStreetMap contributors 2023. Distributed under the Open Data Commons Open Database License (ODbL) v1.0). (b) The enlargement of the Henan province. (c) and (d) are two examples of the new construction of buildings from 2016 to 2021, located at "115.087461ºE, 35.769057ºN" and "113.686257ºE, 34.517389ºN", respectively.**

References:

Gong, P., Li, X., & Zhang, W. (2019). 40-Year (1978–2017) human settlement changes in China reflected by impervious surfaces from satellite remote sensing. Science Bulletin, 64(11), 756–763.

Comment 10: In Line 383, the Radoux et al. (2014) mainly emphasized the spatial heterogeneity, while this part focuses on the temporal consistency, so the reference might be incorrect.

Response:

Thank you for your careful inspection. We agree that the reference to Radoux et al. (2014) is not appropriate for the point we were trying to make. To address this issue, we have revised the text and removed the reference to Radoux et al. (2014) in that sentence, please refer to line 389 to 395 on page 17:

Due to the possible random bias of our method in locating the boundary of building rooftops

(outlined in Sect. 6.2), inconsistencies in identification results over time for the same building may occur. To address this issue, a temporal homogeneity check approach has been implemented. Specifically, it is assumed that a building's state does not undergo continuous change over three successive years. Building upon the method proposed by Li et al, (2015), a $3 \times 3$ sliding window is utilized to determine the final pixel value through majority voting, as illustrated in Fig. 8. This procedure ensures that the results for adjacent years are comparable. However, for the edge year like 2016 and 2021, they are not checked due to the lack of temporal information.

---

## Author Comment (AC3)

**Response to the Anonymous Referee #3**

General comment:

The authors used the STSR-Seg method to develop a novel BRA dataset covering 2016-2021 with a temporal interval one-year, it has been demonstrated to have better performance than Zhang et al. (2022) results and covers the whole China (rural and urban areas), with an overall accuracy of 82.85%. The manuscript has been well-written and clearly organized. However, I still have several concerns about the current manuscript as follows

The authors describe an effort to create building footprint data for all of China. Their dataset is a raster dataset at 2.5m resolution, derived from 10m Sentinel-2 data. They use a deep learning approach involving super-resolution segmentation, allowing to downscale the information from 10m to 2.5m resolution.

The contribution is timely and very relevant, as it tackles several gaps in the global data landscape on human settlements and built-up areas: 1) The created data covers China (including its rural areas), unlike other data products; 2) The dataset is multitemporal (2016-2021) which is rare, allowing for the assessment of built-up growth and shrinkage due to demolition etc.

The paper is well-written and structured. It is very detailed and includes a thorough accuracy assessment against other datasets including a comparison to global datasets available at lower spatial resolution, including datasets from different sources, and also involves hand-crafted validation data and manual checks. The obtained accuracy estimates are quite high and promising.

As I cannot judge the quality of the deep learning framework, I have mostly minor comments, as well as some comments on the data themselves and a request for clarification on the accuracy assessment.

Response:

We are very grateful for the reviewers' comments, which have been very helpful and important in improving the quality of our work. To improve readability, we respond to reviewers' comments in three sections below, i.e., "Part 1: Comments on the data", "Part 2: Accuracy assessment, comparison" and "Part 3: Minor comments".

The reviewer's comments are presented in black font, while our responses are presented in blue font. The revised manuscript is presented in red color, and our modifications are highlighted in yellow.

**Part 1: Comments on the data**

Comments on the data 1:

Empty raster datasets such as CBRA_2016_E113.5_N51.3.tif or CBRA_2016_E76.0_N33.8.tif should be excluded from the dataset.

Response:

Thank you for the reviewer's comment. The empty raster are regions without buildings, they are

now excluded in the final version of the CBRA.

Comments on the data 2:

Until looking at the data, I was very positive towards this manuscript. However, I then had a look at a small selection of the data, and was quite surprised to see very coarse "blobs" delineating settlement areas, rather than mapping "rooftop areas" as the dataset suggests (example in the figure below). I zoomed into 3-4 regions, and most seem to be finer-grained than these blobs and actually delineating individual buildings / rooftops. However, the authors should be transparent and also show such an example in their manuscript, to highlight that the method does not seem to work well everywhere – and possibly provide an explanation for this. Looking at this specific example, I don't think it is defendable to call this "rooftop areas" – this is a quite generalized settlement area, slightly more refined than the GHS-BUILT 10m dataset, shown for comparison.

Response:

Thank you for the reviewer's comment and careful inspection.

Our method employs convolutional neural networks (CNNs) and utilizes super resolution (SR) and semantic segmentation (SS) techniques, which enable us to generate 2.5 m building rooftop results using only 10 m Sentinel-2 images. The following is a simplified diagram of the model architecture (including only the forward inference process).

[Figure]

**Figure C1: A brief structure of the proposed STSR-Seg, mainly including two parts, i.e., super resolution (SR) and semantic segmentation (SS). In our manuscript, the SR is EDSR module and the SS is the U-net module.**

Since the spatial resolution of our results is 2.5 meters, it is difficult to distinguish single buildings when the distance between two buildings is less than 2.5 meters. Furthermore, the deep-learning-based approach we used naturally introduces local ambiguity at the edges of the building, resulting in a buffer-like effect of a few pixels at the building edges (Ding et al., 2021; Zorzi et al., 2021; Guo et al., 2022; Liu et al., 2022). We have included examples of this effect in Fig. C2, where we present an example of boundary localization results using the Unet++ method (Zhou et al., 2019). Even in very high-resolution images (0.3 m), the building boundary extraction results exhibit an uncertain offset of several pixels.

[Figure]

(a)            (b)            (c)            (d)

**Figure C2: An example of the ambiguity of building edges from Inria building dataset (0.3 m) by the Unet++ method (Zhou et al., 2019). (a) The input image (0.3 m). (b) The ground truth. (c) The extraction result. (d) The boundary result, red is ground truth and blue is prediction. We can see even in the very high-resolution images (0.3 m), the building boundary extraction results still shows with an offset of several pixels.**

In densely residential areas, the aforementioned ambiguity is further compounded, as demonstrated in Fig. C3. When buildings stand closely (e.g., less than 6-7 m, i.e., 2-3 pixels at 2.5 m resolution), our method may not be able to effectively distinguish between them. It is a common phenomenon in building extraction task, but the previous work mainly uses very high-resolution images (less than 1 m), making their results seem better than ours.

[Figure]

(a)                        (b)

**Figure C3: Two examples of blob-like areas and the measured distances. (a) Densely residential area (101.302089ºE, 21.298532ºN). (b) Relatively discrete residential area (121.634662 ºE, 31.746674 ºN). Imagery © 2023 Maxar Techonlogies.**

Regarding the potential issue of the inherent ambiguity in edge extraction, we ever conducted an offline experiment to explore other methods that could address this issue, such as RNN-based methods, GAN-based methods, and post-processing methods. We kept the super-resolution module constant and only replaced these methods with our semantic classification head (i.e., the SS part in Fig. C1). However, the results were not satisfactory.

For RNN-based methods, we utilized the Polygon-RNN (Castrejon et al., 2017). However, we

found the model got a trick solution whatever the input, and could not be trained successfully (Fig. C4). For GAN-based methods and post-processing methods, we tried the recently introduced ASLNet (Ding et al., 2021) and FrameField (Girard et al., 2021), and the results were also not good compared with our method. The ASLNet could only obtained 27.64% IoU and the FrameField could only obtain 16.67%, while our method reported in the manuscript is 45.51%. We believe these kinds of methods are all designed for high-resolution building extractions (e.g., less than 1 m), thus showing inferior performance in Sentinel-2 imagery.

[Figure]

(a)  (b)

(c)  (d)

**Figure C4: The result of Polygon-RNN, we replace the SS part in Fig. C1 with Polygon-RNN. (a) and (c) are the ground truth, (b) and (d) are the model prediction results. It could be seen that the method learns a trick solution to the input.**

Finally, we present the possibility map of our method output in Fig. C5. In our manuscript, we used a threshold of 0.5 to binarize this map, following common practice. However, using a higher threshold in areas with dense building clusters may improve the method's performance in these challenging scenarios. We are now working to provide CBRA's building rooftop probability product (to supplement the published data with the original data link) later to serve the need for more precision.

[Figure]

[Figure]

0                                        0.94

**Figure C5: (a) The possibility map of the failure area (101.302089ºE, 21.298532ºN). (b) The possibility map with high-resolution imagery from ArcGIS online. (c) An enlargement of the possibility map. It could be seen that the building rooftop area has a higher possibility (red colour). In our study, we used a threshold of 0.5 to obtain the ultimate binary building rooftop extraction outcome, which is consistent with the standard practice of building extraction. However, increasing the threshold value could potentially improve the results for identifying highly dense buildings in rural regions of China. Imagery © 2023 Maxar Techonlogies.**

To summarize, the ambiguity inherent in edge extraction results in a blob-like shape in our proposed method, with a potential offset of 1-3 pixels. Despite trying other methods specifically designed for accurate building boundary delineation, we found that they were not successful in Sentinel-2 imagery. It should be noted that our method may not be suitable for identifying very dense buildings in rural areas of China, but we are working to provide the probability map in the near future (we will be releasing a new dataset of CBRA probability maps, which is still in production. We will add a link to the probability map product on the CBRA dataset page when it is ready). Despite this limitation, our method is the first to use Sentinel-2 10 m imagery to achieve long-term and large-scale building rooftop mapping at 2.5 m resolution. This low-cost and dynamic building mapping strategy has not been previously achieved.

We now strengthen the statement of limitation (Sect. 6.2) in the revised manuscript (Fig. 19 in the revised manuscript is the Fig. C3 above), from line 608 to 614 on page 30:

Although our STSR-Seg framework is scalable, allowing larger areas to be monitored (e.g., national scale), there remain some limitations in our approach. Specifically, the segmentation results for densely populated residential areas may present certain rooftops as a single block, rather than individual buildings. Our analysis suggests that this occurrence is primarily due to the consequence of the limited spatial resolution of the results, i.e., 2.5 m. Furthermore, the semantic segmentation technique utilized in the approach may introduce some uncertainty at the edges of buildings, resulting in additional pixels, up to three pixels, at the boundary. Consequently, up to 7.5 meters of buffering may occur, exacerbating the problem of building adhesion. Examples of this issue are presented in Fig. 19…….

References:

Ding, L., Tang, H., Liu, Y., Shi, Y., Zhu, X. X., & Bruzzone, L. (2021). Adversarial shape learning for building extraction in VHR remote sensing images. IEEE Transactions on Image Processing, 31, 678–690.

Zorzi, S., Bittner, K., & Fraundorfer, F. (2021). Machine-learned regularization and polygonization of building segmentation masks. 2020 25th International Conference on Pattern Recognition (ICPR), 3098–3105.

Guo, H., Du, B., Zhang, L., & Su, X. (2022). A coarse-to-fine boundary refinement network for building footprint extraction from remote sensing imagery. ISPRS Journal of Photogrammetry and Remote Sensing, 183, 240–252.

Liu, Z., Tang, H., & Huang, W. (2022). Building Outline Delineation From VHR Remote Sensing Images Using the Convolutional Recurrent Neural Network Embedded With Line Segment Information. IEEE Transactions on Geoscience and Remote Sensing, 60, 1–13.

Zhou, Z., Siddiquee, M. M. R., Tajbakhsh, N., & Liang, J. (2019). Unet++: Redesigning skip connections to exploit multiscale features in image segmentation. IEEE Transactions on Medical Imaging, 39(6), 1856–1867.

Castrejon, L., Kundu, K., Urtasun, R., & Fidler, S. (2017). Annotating object instances with a polygon-rnn. Proceedings of the IEEE Conference on Computer Vision and Pattern Recognition, 5230–5238.

Girard, N., Smirnov, D., Solomon, J., & Tarabalka, Y. (2021). Polygonal building extraction by frame field learning. Proceedings of the IEEE/CVF Conference on Computer Vision and Pattern Recognition, 5891–5900.

Comments on the data 3:

I would anticipate much wider usage of the data if they were provided as vector data (i.e. polygon objects describing each building) rather than raster data. The fact that the authors provide 2.5m-raster data still leave a major chunk of processing work to the user. While there are applications where the fine-grained raster data is useful, most applications will be based on vector data. The authors correctly mention the vectorization step as "future work", but I would like to raise the discussion here if it would be beneficial to do this at this point, or otherwise provide a vectorized version of the data in the near future – just as some "food for thought".

Responses:

Thank you for your valuable comment.

We acknowledge the significance of the vector results in building rooftop extraction. However, we have identified limitations in our results, particularly in the segmentation of single buildings in densely populated residential areas, as highlighted in our response to "Comments on the data 2", which is why we decided not to publish the vector version of the CBRA until we can overcome

these limitations and ensure the highest quality of data for users.

However, we understand the significance of vector data for many applications, and we plan to address this in the near future by performing the following tasks:

(1) We will provide the vector version of the CBRA using commonly used vectoring algorithms. This initiative is expected to facilitate more extensive utilization of the data and ease its incorporation into research projects (we will be releasing a new vector version for CBRA, which is still in production. We will add a link to this product on the CBRA dataset page when it is ready).

(2) We will make our source code and training data publicly available after the completion of the review process. This will enable researchers to replicate and build upon our work and encourage the development of new methods that can achieve higher accuracy in building rooftop extraction with relatively coarse resolution imagery (the code will be found in https://github.com/zpl99/STSR-Seg, we have already mentioned in the Introduction of the manuscript).

(3) We plan to develop new methods that can achieve higher accuracy in building rooftop extraction, particularly in densely residential areas.

We now rewrite the Sect. 6.2 "Limitations and prospects" to be more specific, from 615 to 639 on page 30:

……Besides, there is a need for further improvement in the delineation of the building boundaries within the CBRA. Buildings differ from other objects of interest in that they have regularized boundaries (e.g., polygons made of lines and vertexes). However, our dense pixel-to-pixel classification method disregards the morphology of the building, resulting a blob-like shape. For example, in Table 5……This indicates that the CBRA results suffer from ambiguous localization on the building boundaries.

We have noticed that there are many studies on the morphology extraction of buildings in recent years, such as instance segmentation methods (Liu et al., 2022; Zhu et al., 2021; Huang et al., 2021a). We also try to replace our semantic segmentation branch with current instance segmentation methods, e.g., recurrent neural network methods (Liu et al., 2022). However, the results are not good and even fail in our off-line experiment, mainly because these methods are designed for very high-resolution aerial images (sub-metric level). In addition, the efficiency of these methods is too low to support national-level building mapping.

……Many endeavors utilize a post-processing strategy, e.g., Douglas–Peucker algorithm, to achieve regularization (Wei et al., 2019; Chen et al., 2020; Zorzi et al., 2021) and such strategy has shown the success in building mapping at a relatively small scale (Wei et al., 2019). However, in the CBRA, the use of post-processing would introduce errors due to several block estimations in the densely residential area as aforementioned. Considering the potential errors by vectorization, it is hard to provide vector results of the CBRA.

The CBRA provides full-coverage and multi-annual information of building rooftops for China at 2.5 m spatial resolution, and the proposed STSR-Seg offers an opportunity to obtain highresolution output by using relative low-resolution remote sensing images. However, our findings are constrained by the adhesion of closely located buildings and the blob-like shapes of rooftops. In the near future, we aim to enhance our methodology by designing more powerful model architecture and utilizing multisource data, including synthetic-aperture radar (SAR), and other BRA datasets, with the goal of achieving vector outputs.

**Part 2: Accuracy assessment, comparison**

Accuracy assessment, comparison # 1:

Table 4: Why is there only recall reported for the rural scenes, whereas for the urban scenes you report recall, F1, Iou, OA? And why you do not report Precision in both cases? This need to be done and is standard for an accuracy assessment. Of course the reader could calculate the precision based on recall and F1, but please provide Precision, recall, OA, F1, IoU for both the rural and the urban scenario. No rationale is provided for only reporting recall in the rural scenario.

Response:

Thank you for the reviewer's comment.

Regarding our decision to report only recall in the rural scenario, we wish to clarify that this was primarily due to limitations in our test sample selection. Specifically, while we were able to utilize accurate and reliable test samples in urban areas by employing vector data from the National Platform for Common Geospatial Information Service of China, we encountered difficulties in identifying equally reliable validation samples for rural areas.

In order to address this issue, we turned to building distribution data sourced from Open Street Map (OSM) (line 217), which constitutes a form of volunteer geographic information data. However, given the imperfect development of such data in the Chinese region, we recognized that it is subject to errors. To mitigate this, we corrected the OSM data based on the "World Imagery" provided by ArcGIS online, although it should be noted that this imagery does not provide a specific acquisition time (line 218).

Despite these limitations, we made every effort to ensure the accuracy of our existing test samples, and we therefore opted to report recall as the primary metric in rural areas. This was because we could accurately calculate the true positives (TP) and false negatives (FN) based on the existing test samples in the rural area. It should be noted, however, that the calculation of false positives (FP) and true negatives (TN) may be subject to some bias, as certain background pixels which should be classified as building pixels may not have been successfully identified by our visual inspection due to the uncertainty of acquisition time of high-resolution images.

We now add more explanations in the revised manuscript.

In "Section 3.2 Reference data", from 219 to 220 on page 10:

……by ArcGIS Online (Arcgis online, 2022). Despite our efforts, the accuracy of our interpretation is subject to some omissions due to the uncertainty of acquisition time of the images used. Finally, building rooftops of 14 villages are obtained (30, 000 buildings), as shown in Fig.

4…….

Table 4: Performance metrics for building rooftop extraction. Only recall with respect to OSM data is reported in rural areas, due to the challenges of accurately calculating other metrics caused by omissions in the OSM data.

Accuracy assessment, comparison # 2:

Moreover, it is unclear how the accuracy estimates for the Global Human Settlement Layer (GHSL) as reported in Fig. 4 were produced. Which of 10m GHS-BUILT data products was used? There is either the GHS_BUILT_S_E2018_GLOBE_R2022A_54009_10_V1_0 dataset, or the GHS-BUILT-S2 dataset. Both are continuous, with the former reporting the 10m built-up fraction, and the latter reporting the built-up probability. Please provide the following information: Which of the datasets was used? And how were these continuous data thresholded in order to carry out a binary (2-class) agreement assessment? I.e., what cut-off value was used, and how was this cut-off value derived?

Response:

Thank you for the comment.

We used it considering the problem that the threshold is not well delineated. So, the datasets we used is GHS_BUILT_C_MSZ_E2018_GLOBE_R2022A_54009_10_V1_0 product, which provides the category labels for each pixel, as shown in the figure below. For the specific processing, we reclassify this raster data, i.e., 01-05 is assigned to 0 and 11-25 is assigned to 255.

```
01 : MSZ, open spaces, low vegetation surfaces NDVI <= 0.3
02 : MSZ, open spaces, medium vegetation surfaces 0.3 < NDVI <=0.5
03 : MSZ, open spaces, high vegetation surfaces NDVI > 0.5
04 : MSZ, open spaces, water surfaces LAND < 0.5
05 : MSZ, open spaces, road surfaces
11 : MSZ, built spaces, residential, building height <= 3m
12 : MSZ, built spaces, residential, 3m < building height <= 6m
13 : MSZ, built spaces, residential, 6m < building height <= 15m
14 : MSZ, built spaces, residential, 15m < building height <= 30m
15 : MSZ, built spaces, residential, building height > 30m
21 : MSZ, built spaces, non-residential, building height <= 3m
22 : MSZ, built spaces, non-residential, 3m < building height <= 6m
23 : MSZ, built spaces, non-residential, 6m < building height <= 15m
24 : MSZ, built spaces, non-residential, 15m < building height <= 30m
25 : MSZ, built spaces, non-residential, building height > 30m
```

**Figure C6: Morphological Settlement Zone (MSZ) Legend (Schiavina et al., 2022)**

References:

Schiavina, M., Melchiorri, M., Pesaresi, M., Politis, P., Freire, S., Maffenini, L., Florio, P., Ehrlich, D., Goch, K., & Tommasi, P. (2022). GHSL Data Package 2022.

Accuracy assessment, comparison # 3:

The observed accuracy drop from urban towards rural is typical for settlement mapping, please place your work in the context of the literature, e.g. by citing Leyk et al. 2018, or Kaim et al. 2022.

Response:

Thank you for the reviewer's comment. The literature work is now added to the revised manuscript, from line 474 to 480 on page 20:

……Compared to other datasets that provide information related to buildings in rural areas, the CBRA is at a significantly fine-grained scale, albeit with a greater presence of block areas in rural versus urban environments (Fig. 13).

……Besides, the CBRA has a full coverage of China, including the rural areas at a finer scale than other existing full-coverage and thematic-related products. However, a decline in accuracy in rural areas, consistent with prior studies (Leyk et al., 2018; Kaim et al., 2022), has been observed……

References:

Leyk, S., Uhl, J. H., Balk, D., & Jones, B. (2018). Assessing the accuracy of multi-temporal built-up land layers across rural-urban trajectories in the United States. Remote Sensing of Environment, 204, 898–917.

Kaim, D., Ziółkowska, E., Grădinaru, S. R., & Pazúr, R. (2022). Assessing the suitability of urban-oriented land cover products for mapping rural settlements. International Journal of Geographical Information Science, 36(12), 2412–2426.

Accuracy assessment, comparison # 4:

The authors use the overall accuracy for their accuracy assessment. However, it is well-known that OA yields biased results in the case of imbalanced class distributions (see Shao et al. 2019, Uhl & Leyk 2022) for a recent in-depth study. Such class imbalance is typically the case for built-up vs not built-up assessments, in particular in rural areas. Under the light of this potential bias, please add some sentences critically evaluation the magnitude of the OA values obtained. That being said, I appreciate the authors also report IoU and F-1.

Response:

Thank you for the reviewer's suggestion. We now add more descriptions about this issue, from 441 to 444 on page 18:

……For recall, the CBRA obtains 74.66%, which achieves great improvement (+ 27.29%) compared with 90-cities-BRA, mainly because our robust designation of STSR-Seg framework. It is noteworthy that solely relying on OA for evaluating the performance of CBRA is inadequate due to the category-unbalanced nature of building roof extraction. The OA score may introduce a potential bias in this scenario (Shao et al., 2019; Uhl and Leyk, 2022), and therefore, multiple metrics must be utilized when assessing the performance of CBRA.

References:

Shao, G., Tang, L., and Liao, J.: Overselling overall map accuracy misinforms about research reliability, Landsc Ecol, 34, 2487–2492, 2019.

Uhl, J. H. and Leyk, S.: A scale-sensitive framework for the spatially explicit accuracy assessment of binary built-up surface layers, Remote Sens Environ, 279, 113117, 2022.

Accuracy assessment, comparison # 5:

Fig. 16: Legend should be swapped – the blue should be on the left, and red on the right, also in Fig. 18a.

Response:

Thank you for your careful inspection, the Fig. 16 and 18 now are corrected in the revised manuscript.

Accuracy assessment, comparison # 6:

Fig. 17 b and c: I don't understand what is the difference between panel b) and c), besides the different visualization technique. Please clarify. Moreover, I don't think the pie charts are a good choice here. They don't show the change over time. Please use a layer plot for b) just, as you did in c).

Response:

Thank you for your valuable comment.

The purpose of including subfigure (b) was to illustrate the variation in the proportion of rooftop area of buildings in different urban clusters of China, with respect to the total national rooftop area. We acknowledge that this subfigure may appear redundant with subfigure (c). Moreover, the pie chart representation may not be the most appropriate for displaying changes over time.

In response to your feedback, we have integrated subfigure (b) and (c) into a single figure in the form of a stacked bar chart, which is presented below for your review:

[Figure]

(a)

(b)

(c)

**Figure 17: The change of building rooftop area of China and three biggest city clusters in China (NCP, YRD and GBA) over the period of 2016-2021. (a) The annual statistic of building rooftop area in China. (b) The proportion of building rooftop of the biggest city clusters in China and other regions from 2016 to 2021. (c) The increased building rooftop area on each city clusters and other regions.**

Accuracy assessment, comparison # 7:

Fig. 18: the green color used to show demolition is different in the map and in the legend.

Response: Thank you for your careful inspection, the 18 now is updated with consistent color in the revised manuscript.

Accuracy assessment, comparison # 8:

Fig. A2, caption: Figure A2: The probability density distribution. …. Of what???

Response: Thank you for your comment. It should be the probability density distribution of the ratio of buildings to built-up area. We compute the probability density distribution of this ratio and observe that it approximates a Gaussian distribution. Leveraging this prior knowledge, we train the model to extend its coverage to areas where high-resolution samples are unavailable. We now add more details to the caption of Fig. A3 in the revised manuscript.

**Part 3: Minor comments**

Minor comments #1: Please provide a rationale for using the term "rooftop area" instead of "building footprint area" or "built-up area"

Response:

Thank you for the reviewer's suggestion.

In our research, we aim to extract individual buildings from Sentinel-2 satellite images using super-resolution techniques which help enhance the original resolution of 10 m to a new resolution of 2.5 m. We appreciate the suggestion to consider the use of alternative terms such as "built-up area" or "building footprint area."

The term "built-up area" refers to the total area of land that has been developed or modified by human activity, including buildings, roads, and other infrastructure. However, as our research focuses on the specific features of individual buildings rather than the broader urban environment, we found that the term "built-up area" was not appropriate for our study.

"Building footprint" refers to the total area that a building covers on the ground, including any exterior walls or other structural elements that extend beyond the building's primary enclosed space. While this term is useful for understanding the physical layout of a building, we found it challenging to directly extract building footprints from Sentinel-2 data due to resolution limitations. As an alternative, we chose to use the term "rooftop area," which specifically refers to the top surface of a building and provides a more practical option for our research.

We acknowledge that both "building footprint area" and "rooftop area" are specific to individual buildings and may be appropriate in different contexts. However, in our research, we found that "rooftop area" was a suitable term to describe our focus.

Minor comments #2: Line 75: What means "and F1 score of 2.5 m," .... I don't understand what the authors mean here.

Response:

Thank you for the reviewer's comment. We originally meant to emphasize that the resolution of our results is 2.5 meters, but this sentence is redundant. We now remove the word "2.5 m" in the revised manuscript.

Minor comments #3: Line 106: No need to define an acronym for state-of-the-art; the term is only used twice in the paper.

Response:

Thank you for the reviewer's suggestion. We remove this acronym in the revised manuscript.

Minor comments #4: Line 159: "apple" ?

Response:

We apologize that this is a spelling error on our manuscript, it should be "apply". We now correct it in the revised manuscript.

Minor comments #5: Line 161: Please explain what you mean by "geographical offset".

Response:

Thank you for your inquiry.

The geographical offset we refer to is the bias in the GES imagery. This bias is mainly caused by the following reason:

The acquisition of larger scale GES images generally necessitates the stitching of images from multiple sensors. However, this process can result in large errors at the stitching regions, especially in areas with significant height variations, such as high-rise buildings. Examples of such errors are presented in Fig. C7, which were obtained using Google Earth Pro software.

[Figure]

(a)                                                    (b)

Figure C7: Two examples about the stitching part of GES imagery. (a) The high-rise buildings. (b) The low-rise buildings. It could be seen that the offset is more obvious in the high-rise buildings than that in the low-rises. Imagery © 2023 Maxar Techonlogies.

Details regarding this phenomenon are introduced in the revised manuscript (Fig. A1 is the Fig. C7 above), from line 159 to 160 on page 6 :

However, the GES images are collected from various kinds of high-resolution satellites, and have two potential problems when applied to large-scale mapping: (1) inconsistent geographical offset (illustrated in Fig. A1), and (2) inconsistent acquisition time (e.g., the image is obtained from various satellite sensors with different acquisition times) which results in spatio-temporal inconsistency in the generated product.

Minor comments #6: 155-165: nice transition and justification for the contribution of the paper.

Response:

Thank you for your comment.

Minor comments #7: Table 1. Nice overview on existing datasets.

Response:

Thank you for your comment.

Minor comments #8: Fig. 2: Please include the GHS-BUILT dataset here, from the Global Human Settlement Layer, e.g., the GHS-BUILT-10m built up layer. This will provide a nice overview on recent work at a global scale, and highlight the merit of your work.

Response:

Thank you for your comment. The GHS-BUILT-10m built up layer is now added to Fig. 2 on page 6:

[Figure]

**Figure 2: An example of the result from several representative building-related datasets (121.344419ºE,31.093870ºN). The GAIA (Gong et al., 2020b) reflects the impervious area (30 m). The WSF (Marconcini et al., 2020b) and GHSL (Corbane et al., 2021) are the human settlement data (10 m). The CBRA (ours) is the building rooftop area data (2.5 m).**

Minor comments #9: Caption Fig. 2: "cloud under" change to "cloud cover under"

Response:

Thank you for your comment. Correct.

Minor comments #10: Fig. 4: Text is very small, please increase font size, and decrease spacing between lines; this way, the space can be used more efficiently.

Response:

Thank you for your suggestion. Fig. 4 is updated in the revised manuscript, on page 9:

[Figure]

**Figure 4: Illustration of the collected high-resolution reference. (a) is the high-resolution reference distribution map (base map © OpenStreetMap contributors 2023. Distributed under the Open Data Commons Open Database License (ODbL) v1.0). (b) and (c) are real-world examples of the collected survey data (43.88 °N, 125.37 °E) (survey data © Tiandi-Map) and the volunteered data (33.47 °N, 119.79 °E) (volunteered data © OpenStreetMap contributors 2023. Distributed under the Open Data Commons Open Database License (ODbL) v1.0), respectively. (d) is the statistic of building rooftops.**

Minor comments #11: Fig. 9 – caption: "Comparison of the CBRA and the other dataset" – please name the "other dataset".

Response:

Thank you for your comment. We now replace "other dataset" with "90-cities-BRA (Zhang et al., 2022)"

References:

Zhang, Z., Qian, Z., Zhong, T., Chen, M., Zhang, K., Yang, Y., Zhu, R., Zhang, F., Zhang, H., & Zhou, F. (2022). Vectorized rooftop area data for 90 cities in China. Scientific Data, 9(1), 1–12.

---

## Referee Report (RR1)

Example 1

[Figure]

Example 2

[Figure]

Example 3